# On the Highly Ordered Graphene Structure of Non-Graphitic Carbons (NGCs)—A Wide-Angle Neutron Scattering (WANS) Study

Oliver Osswald [1],*, Marc O. Loeh [2], Felix M. Badaczewski [2], Torben Pfaff [3], Henry E. Fischer [4], Alexandra Franz [5], Jens-Uwe Hoffmann [5], Manfred Reehuis [5], Peter J. Klar [6] and Bernd M. Smarsly [1],*

1   Institute of Physical Chemistry, Justus-Liebig-University Giessen, Heinrich-Buff-Ring 17, 35392 Giessen, Germany
2   Schunk Kohlenstofftechnik GmbH, Rodheimer Strasse 59, 35452 Heuchelheim, Germany
3   Lang GmbH & Co. KG, Dillstrasse 4, 35625 Hüttenberg, Germany
4   Institut Laue-Langevin, 71 Avenue des Martyrs, CS 20156, CEDEX 9, 38042 Grenoble, France
5   Helmholtz-Zentrum Berlin für Materialien und Energie, Hahn-Meitner-Platz 1, 14109 Berlin, Germany
6   Institute of Experimental Physics I, Justus-Liebig-University Giessen, Heinrich-Buff-Ring 16, 35392 Giessen, Germany
*   Correspondence: carbon@oss-wald.de (O.O.); bernd.smarsly@phys.chemie.uni-giessen.de (B.M.S.)

**Abstract:** Non-graphitic carbons (NGCs), such as glass-like carbons, pitch cokes, and activated carbon consist of small graphene layer building stacks arranged in a turbostratic order. Both structure features, including the single graphene sheets as well as the stacks, possess structural disorder, which can be determined using wide-angle X-ray or neutron scattering (WAXS/WANS). Even if WANS data of NGCs have already been extensively reported and evaluated in different studies, there are still open questions with regard to their validation with WAXS, which is usually used for routine characterization. In particular, using WAXS for the damping of the atomic form factor and the limited measured range prevent the analysis of higher-ordered reflections, which are crucial for determining the stack/layer size ($L_a$, $L_c$) and disorder ($\sigma_1$, $\sigma_3$) based on the reflection widths. Therefore, in this study, powder WANS was performed on three types of carbon materials (glass-like carbon made out of a phenol-formaldehyde resin (PF-R), a mesophase pitch (MP), and a low softening-point pitch (LSPP)) using a beamline at ILL in Grenoble, providing a small wavelength and thus generating WANS data covering a large range of scattering vectors (0.052 Å$^{-1}$ < *s* < 3.76 Å$^{-1}$). Merging these WANS data with WANS data from previous studies, possessing high resolution in the small *s* range, on the same materials allowed us to determine both the interlayer and the interlayer structure as accurately as possible. As a main conclusion, we found that the structural disorder of the graphene layers themselves was significantly smaller than previously assumed.

**Keywords:** non-graphitic carbon; wide-angle scattering; disorder-determination

## 1. Introduction

Carbon occurs in many forms, of which diamonds (made up of $sp^3$ hybridized carbon) and graphite (made up of $sp^2$-hybridized carbon) are the most common ones. Among the $sp^2$-hybridized carbon materials, the so-called "non-graphitic carbons (NGCs)" represent a million-ton-scale class and are of significant relevance for applications. NGCs comprise a plurality of carbons, such as activated carbon, glass-like carbon, and bio chars such as charcoal. As a bulk material, it can be used for different electrical and low friction applications [1–3], whereas the porous derivates and the carbide derived carbons (CDCs) are commonly used in gas storage/separation, e.g., in carbon molecular sieves (CMS) [4–8], as electrodes in sulfur lithium batteries [6,9,10] and super capacitors [11–13], and as catalyst supports in different syntheses [14,15]. Also the so called "glass-like" carbons, which are

available from phenol formaldehyde (PF) precursors, are part of the NGC family and are commercially used due to their excellent thermal and chemical stability [16], e.g., to produce carbon fiber reenforced carbons for high-temperature applications [17].

In general, NGCs consist of sp2-hybridized (graphene) layers (Figure 1), which are stacked parallel on top of each other, but are randomly arranged by rotation perpendicularly to the layer ("turbostratic" structure), as already proposed by Warren [18]. The graphenes and the stacks have nanometer dimensions and possess substantial structural disorder, both in the graphenes themselves as well as in the stacking, i.e., the stacking distance between adjacent graphenes exhibits a broad distribution (Figure 1). This absence of long-range crystallographic order causes broad and overlapping scattering maxima in WAXS and WANS. It is important to note that this feature constitutes the definition of NGCs by the IUPAC [19]. The main structural dimensions of these stacks are described by the parameters $L_a$ and $L_c$, which are the average lateral extension and the stack height, respectively. Further important and relevant parameters are the C-C bond length ($l_{cc}$), the average distance $\overline{a_3}$ between the layers, and parameters quantifying the substantial degree of disorder (strain) within the layers themselves and their stacking ($\sigma_1$ and $\sigma_3$).

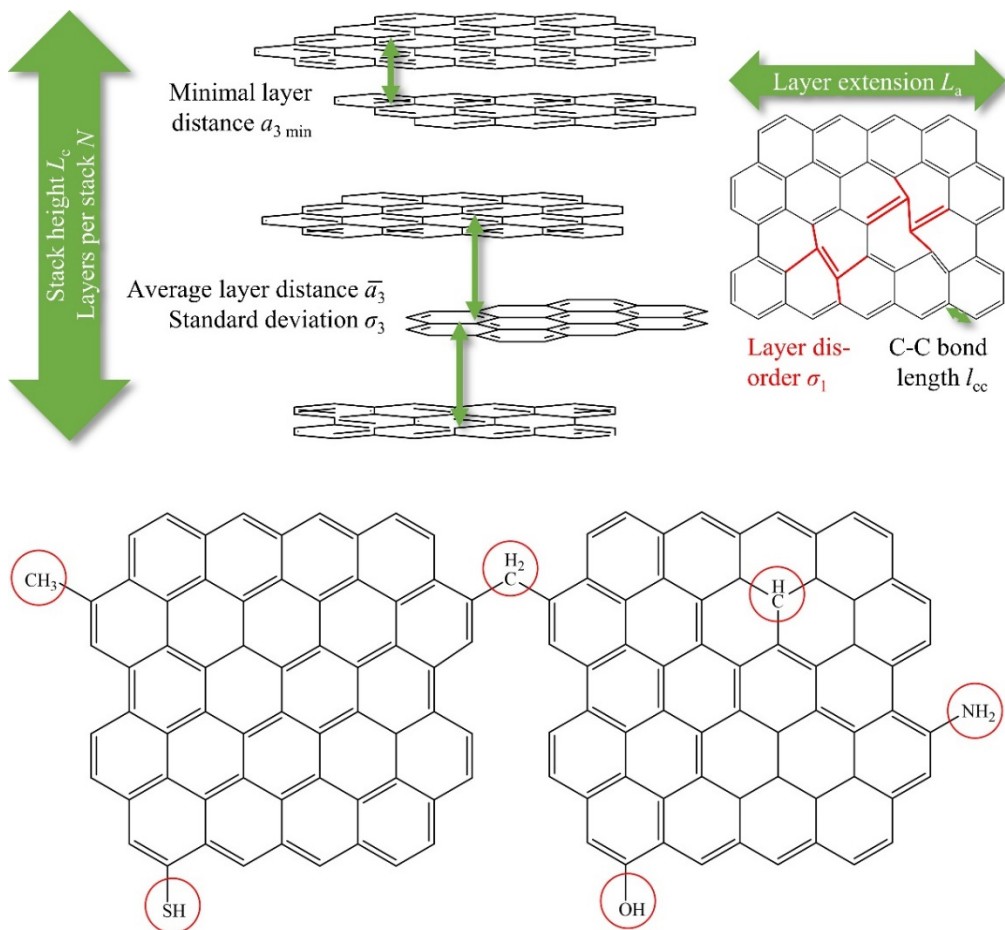

**Figure 1.** Principle structure of non-graphitic carbons (NGCs) containing a turbostratic stacking arrangement of single (disordered) graphene layers. Also, NGCs can contain unorganized carbon ($sp^3$-hybridized) or foreign atoms (e.g., H, N, O, S). Adapted from [20,21].

Given the broad industrial application of NGCs, a quantitative determination of the aforementioned microstructure based on experimentally accessible structural parameters is crucial for the tuning of production processes and fundamental understanding of the linkage between the microstructure and macroscopic material properties. The latter comprises

hardness, chemical stability, thermal properties, and electrical conductivity, which hence defines the final application.

Three main approaches are usually applied for the microstructural characterization of NGCs, which are transmission electron microscopy (TEM) [22,23], Raman scattering [24–26], and wide-angle X-ray scattering (WAXS), is the latter being used as one of the first methods to characterize such carbons [25,27–30]. Among them, WAXS offers fundamental advantages, particularly in the straightforward experimental analysis using standard X-ray powder diffraction laboratory setups. Raman scattering is used as a routine method that provides the lateral extension of the graphene layers and a qualitative overview on structural order, but is especially useful for small graphene dimensions and disordered layers, yet the analysis needs further validation with respect to other structural features [25]. Instead, WAXS or wide-angle neutron scattering (WANS) can provide a larger number of structural parameters, especially regarding structural disorder. The main features of WAXS/WANS data are the superposition of the scattering from the single graphenes ("intralayer scattering") and their stacking ("interlayer scattering"), as well as the large width of the corresponding reflections (see Figure 2). It is important to emphasize that the two essential structural features of NGCs, the graphenes and the stacks, both possess a small dimension of a few nm, as well as structural disorder. Understanding the changes in the graphene dimension and the disorder upon chemical or thermal treatment are, however, desirable and crucial in unraveling the fundamentals of carbonization and graphitization. Importantly, the nanometer-sized dimension and the structural disorder both result in a significant broadening of the interlayer and intralayer reflections. It is thus challenging to separate these effects from WAXS/WANS data to obtain accurate and reliable values for the size-related parameters ($L_a$, $L_c$) and their respective disorders ($\sigma_1$ & $\sigma_3$). Since finite crystallite size and disorder result in a different dependence of the width of reflections with increasing modulus of the scattering vector $s$ ($s = 2\sin(\theta)/\lambda$), the most suitable strategy to disentangle size and disorder is the acquisition of as many reflections as possible, which is equivalent to acquiring WAXS/WANS data up to a large $s$. Unraveling the exact degree of disorder of the graphenes and their stacking, which are quantified by the parameters $\sigma_1$ and $\sigma_3$, constitutes the main motivation of this present study.

However, owing to the broad, overlapping, and asymmetrical reflections, classic evaluation approaches based on analyzing the width of separated WAXS/WANS maxima are inappropriate for the microstructural analysis of NGCs based on WAXS/WANS. In contrast, it is recommended to fit the entire WAXS/WANS curve using a suitable model function. A widely used approach for WAXS was proposed by Shi et al. in 1993 [31], followed by other procedures by Azuma in 1998 [32] and Fujimoto and Shiraishi in 2001 [33]. Finally, in 2002, Ruland and Smarsly published an advanced approach for analyzing the WAXS of NGCs [34] that has already been successfully used in several studies [20,35–42]. Typical WAXS/WANS data and the general idea of the approach of Ruland and Smarsly are shown in Figure 2.

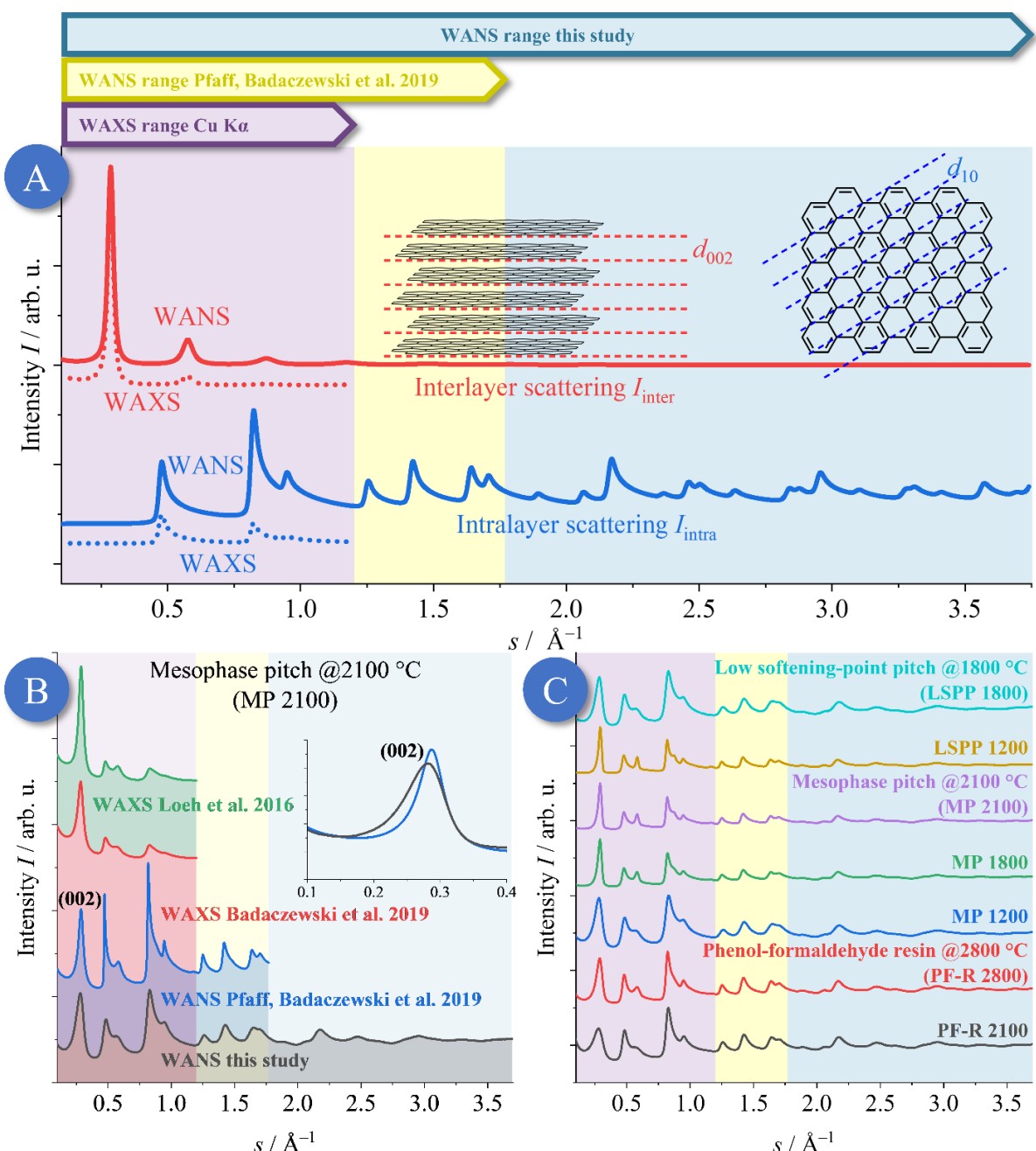

**Figure 2.** (**A**): Simulation of WAXS/WANS data of a representative NGC (without background scattering), which is given by a superposition of symmetric interlayer reflections (00*l*) and asymmetric intralayer reflections (*hk*). By using neutron radiation instead of X-rays, much more intralayer reflections can be measured, and, therefore, the intralayer graphene structure in particular can be determined more accurately. (**B**): One particular sample (mesophase pitch; MP @2100 °C, i.e., treated at 2100 °C) was studied four times, namely using WAXS measurements in the study of Loeh et al. (2016) [40], WAXS measurements in the work of Badaczewski et al. (2019) [37], WANS analysis performed by Pfaff et al. (2019) [43], and WANS measurements, the latter of which were performed in this study. In the present WANS study, a much lower wavelength was used compared to Badaczewski et al. (2019) [43]. While both WAXS measurements are very similar, there are differences between the two WANS measurements (performed at the Berlin and Grenoble facilities) due to the different instrumental resolutions. (**C**): Overview of the WANS data of samples, which were measured two times (2019 and in this study) and combined to get one merged WANS curve with high *s*-space

resolution, and also an extended *s*-range. Here, *s* means the modulus of the scattering vector $s$ ($s = 2 \sin(\theta)/\lambda$). It should be noted that the indexing shown with parentheses strictly denotes lattice planes, and reflections would have to be indexed without parentheses. However, brackets are used in this publication for both meanings to improve readability.

While evaluation approaches such as the one of Ruland and Smarsly provide meaningful structural characterization, the analysis of NGCs by WAXS and thus their validity generally suffers from the following features (Figure 3):

1.  Atomic form factors result in the damping of WAXS data at larger *s* values. This effect is significant for typical XRD setups using a $CuK_\alpha$ wavelength.
2.  The Compton scattering has to be exactly taken into account, but has a complex dependence on *s*.
3.  The limited range of *s* ($s = 2/\lambda \sin(\theta)$) commonly available in standard lab XRD setups reduces the number of accessible WAXS reflections. A large number of reflections is, however, crucial to quantitatively disentangle size ($L_a$ and $L_c$) and disorder effects ($\sigma_1$ and $\sigma_3$ in our approach), which both influence the significant width of the NGC reflections.

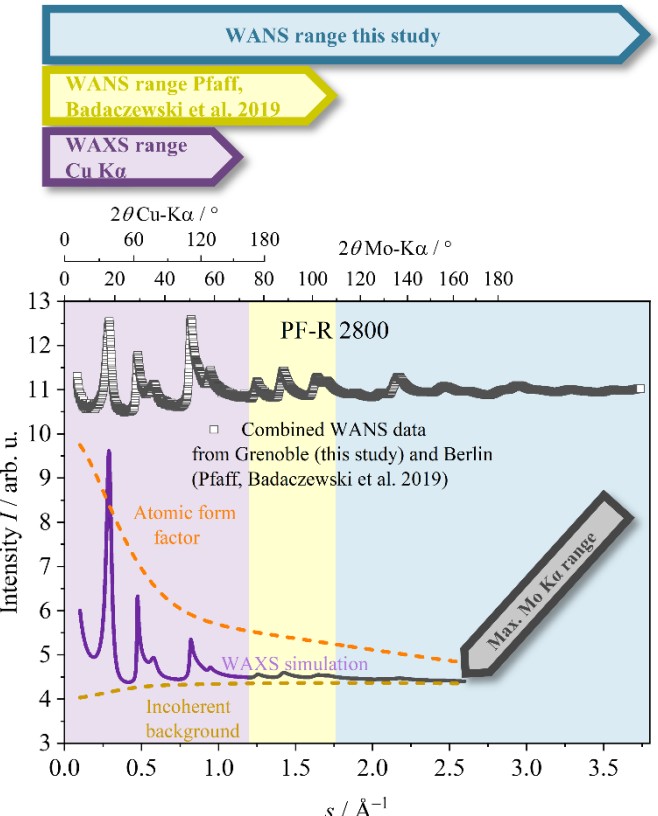

**Figure 3.** Illustration of the shortcomings of X-rays in the analysis of NGCs and the advantage of WANS. The merged WANS data shown (sample PF-R 2800, a phenol-formaldehyde resin) were obtained from a beamtime at Grenoble (this study) and from an HZB (Berlin) beamtime, see Pfaff et al. [43]. While WAXS data suffer from the damping induced by the atomic form factor, by using WANS data more distinguishable reflections can be obtained and analyzed. Also, WANS measurements with, e.g., Mo-K$\alpha$ radiation, which makes high values of the *s* accessible too, may not significantly improve the evaluation, since the atomic form factor leads to a substantial damping in this case. Additionally, when using X-ray radiation, the incoherent background must be taken into account and calculated exactly for a quantitative analysis.

Thus, WANS performed up to large $s$ values provides fundamental advantages in the study of NGCs when compared to WAXS. Recently, our experiments performed at the Flat-Cone Diffractometer (E2 at BER II, HZB, Berlin) [44] revealed that, with a few specific modifications, the approach by Ruland and Smarsly can indeed be applied to the WANS data of NGCs. However, due to wavelength restrictions, the observable range of $s$ was not significantly larger compared to typical lab WAXS experiments. Hence, the full potential of WANS (see points 1–3 above) was not accessible. In particular, the accurate determination of the disorder parameter remained unsatisfactory.

Thus, in the present study WANS data were collected at the D4 wide-angle neutron diffractometer at the Institute Laue–Langevin (ILL) in Grenoble on the very same samples [45,46], complementing the HZB data, based on these considerations:

1.  While the WANS data acquired in our previous study featured high precision particularly at low $s$ (Berlin; Pfaff et al., 2019 [43]), the facility at Grenoble featured advantageous resolution at high $s$ ($0.052 \text{ Å}^{-1} < s < 3.76 \text{ Å}^{-1}$). However, the Berlin HZB WANS data ($0.400 \text{ Å}^{-1} < s < 1.44 \text{ Å}^{-1}$) possessed better resolution in the region of the (002) reflection compared to the WANS data acquired in Grenoble. Thus, combining both data sets for the very same sample was intended to test if a valid deconvolution of the effects of finite size and disorder in both the interlayer and intralayer structure was possible.
2.  The noise of the Grenoble WANS beam line was very low; therefore, the refinement quality and the resulting microstructure parameters were superior. In particular, the analysis of the higher-order reflections, possessing decreasing intensity, benefited from a low background and noise.
3.  The high neutron flux available at the D4 in Grenoble allowed for the acquisition of high-quality data for a substantial number of materials. In particular, in this study, extensive temperature series were possible for materials exposed to a lot of temperature steps. Together with the high data quality, fine changes in the evolution of the microstructure upon temperature treatment were thus accessible.

Based on performing WANS experiments at the Grenoble facility and merging them with the HZB WANS data, which were performed on the very same NGC materials, we therefore aimed at a precise quantification and, thus, interpretation of the disorder of the graphenes and their stacking. Moreover, by applying the quite basic structural model underlying the approach of Ruland and Smarsly, we addressed the question as to whether the concomitant refinement approach is indeed able to fit WANS data up to quite large $s$ values, in the light of the relatively small number of relevant structural parameters ($L_a$, $L_c$, $\sigma_1$, $\sigma_3$) which also act as refinement parameters in this fitting approach. Moreover, the structural analysis was compared with Raman scattering analysis, especially using a novel approach by Schüpfer et al. [25], which could thus result in the verification of the state-of-the-art analysis in this field as well, as identical samples were compared [24–26]. Further, the results were compared to elemental analysis. Based on this method validation, our study is intended to gain further insights into the evaluation of the NGC graphene structure upon temperature treatment up to temperatures that are close to the onset of graphitization.

## 2. Materials and Methods

### 2.1. Sample Preparation

For our studies, we used different NGCs made from different precursors and that were heat treated at different temperatures from 1000 °C to 3000 °C. The precursors were mesophase pitches (MP), low softening point pitches (LSPP/WP) and phenol-formaldehyde resole resins (PF-R). The different precursors were heat-treated to different temperatures between 1000 °C and 3000 °C in an inert gas atmosphere. Each sample was kept for 2 h at the given temperature. The resin was cured in ambient atmosphere at different temperatures (70 °C, 140 °C, 220 °C) for 12 h. After ball milling and washing with deionized water (for resin), all samples (including the pitches) were carbonized in nitrogen atmosphere at 800 °C

for 2 h using a heating rate of 240 °C/h. Additionally, further graphitization (1000 °C up to 3000 °C) was done using a heating rate of 300 °C/h and a residence time of 2 h. The heat treatment at 3000 °C was carried out using an Acheson furnace. Note that the WANS experiments were performed at room temperature.

The samples in Table 1 were measured and evaluated:

**Table 1.** Overview of the refinement of simulated WAXS data, which were blurred by statistical noise generated by a Gaussian distribution. The results indicate that *OctCarb* can evaluate reproducibly microstructural parameters from given WAXS and WANS data. Since the influence of $\sigma_1$ on WAXS (for typical Cu-K$_\alpha$ lab setups) and the influence of $\sigma_3$ on WANS data is small, these parameters cannot always be determined exactly. Therefore, the resulting values for $\sigma_1$ (WAXS) and $\sigma_3$ (WANS) deviated significantly from the input value.

| Heat Treatment Temperature | Long Name | Short Name | WANS This Study | WAXS Ref. [40] | WAXS Ref. [37] | WANS Ref. [43] |
|---|---|---|---|---|---|---|
| | | Phenol-formaldehyde resin (PF-R) | | | | |
| 1000 °C | PF-R 1000 | H 1000 | X | - | X | - |
| 1200 °C | PF-R 1200 | H 1200 | X | - | X | - |
| 1500 °C | PF-R 1500 | H 1500 | X | - | X | - |
| 1800 °C | PF-R 1800 | H 1800 | X | - | X | - |
| 2100 °C | PF-R 2100 | H 2100 | X | - | X | X |
| 2300 °C | PF-R 2300 | H 1300 | X | - | X | - |
| 2800 °C | PF-R 2800 | H 2800 | X | - | X | X |
| 3000 °C | PF-R 3000 | H 3000 | X | - | X | - |
| | | Mesophase pitch (MP) | | | | |
| 1200 °C | MP 1200 | MP 1200 | X | X | X | X |
| 1500 °C | MP 1500 | MP 1500 | X | X | X | - |
| 1800 °C | MP 1800 | MP 1800 | X | X | X | X |
| 2100 °C | MP 2100 | MP 2100 | X | X | X | X |
| | | Low softening-point pitch (LSPP) | | | | |
| 1200 °C | LSPP 1200 | WP 1200 | X | X | - | X |
| 1800 °C | LSPP 1800 | WP 1800 | X | - | - | X |
| 2500 °C | LSPP 2500 | WP 2500 | X | - | - | X |
| 2800 °C | LSPP 2800 | WP 2800 | X | - | - | X |
| 3000 °C | LSPP 3000 | WP 3000 | X | - | X | X |

## 2.2. Wide-Angle Scattering

Wide-angle X-ray scattering was measured using a PANalytical X'Pert Pro powder diffractometer at a wavelength of 1.5418 Å (Cu-K$\alpha$ radiation). As sample holder, a no-background silicon crystal was used. Wide-angle neutron scattering was carried out at the D4 disordered materials diffractometer at the Institut Laue–Langevin (ILL) in Grenoble [46]. The samples were loaded in an approx. 6mm diameter cylindrical vanadium cell at ambient conditions. Three identical cells were used to collect the data as efficiently as possible. The monochromator generated wavelength was about 0.5 Å (refined $\lambda$ = 0.4975 Å), which generated a neutron flux about $5.0 \times 10^7$ cm$^{-2}$ s$^{-1}$ and a quite high $s_{max}$ = 3.76 Å$^{-1}$ ($s_{min}$ = 0.05 Å$^{-1}$) with a resolution of about $\Delta s_{min}$ = 0.005 Å$^{-1}$. The data collecting time for all samples was approx. 100 min. The measurements [45,46] were performed at room temperature and in vacuum with $p < 10^{-3}$ mbar. Measurements were performed on the empty cells, the instrument background, a standard vanadium sample for intensity normalization, and a nickel powder sample for wavelength calibration. The CORRECT program was then used to normalize the data to arns/str/atom, as well as to make attenuation (i.e., absorption) and multiple-scattering corrections.

Background correction was done in a similar way to the method of Osswald and Smarsly [21]. For five samples containing a significant amount of hydrogen ($c$(H) > 0.2%), a

pseudo-Voigt correction was used as described by Fischer et al. [47] and in 2.2 (PF-R 1000, 1200, 1500 and MP 1200, 1500). For all other samples, a Placzek correction [48] was used to determine the background scattering in a meaningful way. Since the measured $s$-range is quite high, it can be assumed that the scattering curve oscillates around a constant value and, to be more precise, the intensity must oscillate around $I = 1$ in order to calculate the scattering intensity $S$ of the $q$ ($S(q)$) curve ($q = 2\pi s$), from which the pair distribution function ($PDF(r)$) can be calculated by a Fourier transform (Equation (1)) [49–53]. This pair distribution function was calculated but will be presented in detail in a following work.

$$PDF(r) = 2/\pi \int_{0}^{q_{max}} q\ (S(q) - 1) \sin(q\ r) \mathrm{d}q \tag{1}$$

*2.3. Elemental Analysis*

A Vario EL from Elementar was used to determine the carbon and hydrogen contents, whereas for oxygen and nitrogen, an Eltra OHN2000 was used. The results are shown in Section 3.5).

*2.4. Raman Spectroscopy*

For the PF-R and LSPP series, the data from Schüpfer et al. in 2020/2021 [24,25] were used. These data were measured with a Renishaw inVia Raman microscope system using backscattering geometry at ambient conditions with a 532 nm laser and a 50× objective. For the PF-R, the integral exposure time was 3 accumulations of 30 s with a laser power of 0.6 mW, using a resolution of ~1.5 cm$^{-1}$ and a range of 1000–3200 cm$^{-1}$. For the LSPP, a 1.5 mW laser was used with an exposure time of 30 s (20 s for LSPP 2500). A total of 10 accumulations were performed for the LSPP 1200 and 3000, 7 accumulations for the LSPP 1800 and 2800, and 15 accumulations for the LSPP 2500. The MP series was measured by a Senterra Raman microscope system from Bruker with 200 accumulations of 5 s in a range of 1000–3200 cm$^{-1}$ using a 50× objective and a wavelength of 532 nm with a 2 mW laser under ambient conditions.

*2.5. Data Treatment*

2.5.1. Background Correction

For the background correction of the WANS data, the models of Placzek [48] or Fischer et al. [47] should be used, where the background scattering can be calculated by fitting by a cubic polynomial or a pseudo-Voigt function. The samples treated at moderate temperatures, i.e., PF-R 1000 and PF-R 1200, contain a significant concentration of hydrogen, which leads to a non-constant background scattering in the WANS data, especially due to hydrogen (see Section 3.5), which possesses a high incoherent scattering cross section. This pronounced non-constant background scattering can dominate the WANS data. Details about these corrections will also be given in the SI file in S1.

2.5.2. WANS Data Combination

Generally, the interlayer parameters were mainly determined using the (002) and the (004) reflections. While the (002) reflection was clearly visible, the (004) reflection was broad and often appeared as only a shoulder of the (10) reflection, and, therefore, it was difficult to determine $L_c$ and $\sigma_3$ values accurately. By using WANS measurements, the (006) reflection can also be used for the analysis because of the absence of damping by a non-constant atomic form factor. However, the instrumental resolution in the $s$-space ($\Delta s/s$) for the present (Grenoble) WANS data was not as good as in our previous experiments at HZB (Berlin). Therefore, the WANS data from Pfaff et al. measured at the HZB in Berlin were combined with the WANS data measured at Grenoble (Figure 4). This led to merged WANS data with high resolution for small (Berlin) as well as high (Grenoble) $s$-values and, therefore, both inter- and intralayer parameters could be determined with high accuracy. A

more detailed description, regarding how this combination was performed, can be found in the SI file under S3.

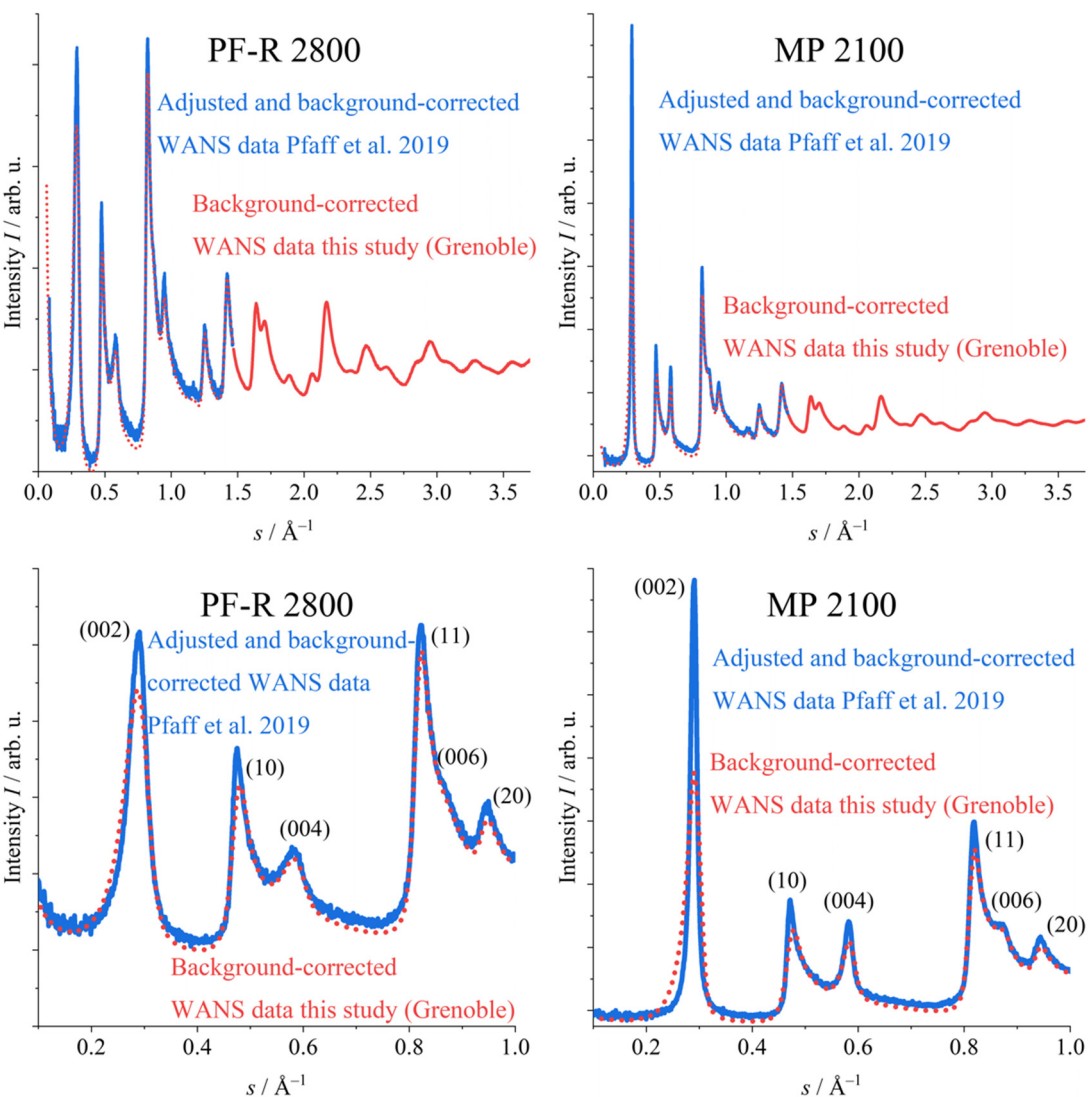

**Figure 4.** The WANS data from Pfaff et al. (2019) [43] (blue) were combined with WANS data from this study (red). In the range of lower $s$-values, the WANS data from this study (red dotted) were only slightly different for PF-R 2800 (disordered glass-like carbon). For MP 2100 (graphitizable mesophase pitch), the differences were more pronounced, since this sample was more ordered and showed sharper reflections. In addition, the $s$-space resolution of the WANS data acquired at the Grenoble facility was lower than that of Pfaff et al. As a result, the reflections in the anterior $s$-region were broader and had lower intensity. Overall, by combining the different WANS data, both the interlayer and intralayer parameters could be accurately determined.

In total, for the 7 samples, WANS data acquired from two different facilities in each of the cases were measured and combined (PF-R 2100/2800, MP 1200/1800/2100, LSPP 1200/1800). Figure 4 shows the combination of the two different samples, namely, the non-graphitizable phenol-formaldehyde resin heat-treated at 2800 °C and the graphitizable mesophase pitch heat-treated at 2100 °C. The significant width of the (002) reflection indicated a quite disordered stacking for PF-R 2800. For MP 2100, the width of the (002) reflection was much smaller, and therefore, the experimental resolution ($\Delta s/s$) exhibited a higher influence on this reflection. The WANS data from Pfaff et al. measured in Berlin [43] did not suffer from such a broadening, and therefore the two sets of WANS data were

combined to get as reliable an analysis as possible. In the following, the figures and tables (as well as the SI file) show both the results obtained from the combined and from the original WANS data of this present study (Grenoble facility).

2.5.3. WANS Data Refinement

The results of the data refinement using the algorithm of Ruland and Smarsly [34] and the Octave refinement script (*OctCarb*) of Osswald and Smarsly [21] are shown in Section 3.2 (only PF-R), Table S1 (all samples) and Figures S1–S4 (comparison to previous studies). The figures contain the most relevant parameters obtained in this study and our previous ones [37,40,43]. A detailed comparison for each structural parameter is given in Tables S3–S35.

For LSPPs 2500/2800/3000, i.e., for high treatment temperatures, mixed (*hkl*) reflections, e.g., (101) and (102) at $s = 0.493$ Å$^{-1}$ and $s = 0.557$ Å$^{-1}$, respectively, were visible (Figure S7 in the SI). Therefore, the approach by Ruland and Smarsly [34] could not be used for these samples. Instead, the Scherrer analysis [54] was applied based on the full width at half maximum (*FWHM*), as described by ref. [24,37,40,41,43] for such materials.

The basic equation used was the following:

$$L_c = \frac{0.93}{FWHM} \frac{\lambda}{\cos(\theta_{center})} \qquad (2)$$

It was used for the background-corrected WANS data for (002) reflections for some of the LSPP samples (2.500 °C and above) using the FWHM of a Gauss-type profile, which is meaningful for such powder diffraction experiments [43,55]. Here, 0.93 was chosen as factor, as described to be meaningful for small and disordered stacks [56]. The $\theta_{center}$ is the position of the Bragg signal in units of radians. The parameter $L_c$ was calculated using Equation (2), and $\overline{a_3}$ and $l_{cc}$ were calculated directly from the positions of the (002) and (10) reflections, respectively, in this case. Additionally, it must be assumed that no further disorder was present in the samples, i.e., $\sigma_3 = 0$ Å.

## 3. Results

### 3.1. Qualitative Discussion of the WANS Data

The background-corrected WANS data of the NGCs from the phenol-formaldehyde resin (PF-R) are shown in Figure 5, and the corresponding data for the low-softening point pitch (LSPP) and mesophase pitch (MP) samples can be found in Figures S6 and S8, respectively. The WAXS/WANS of NGCs is the superposition of interlayer scattering caused by the stacking, and the intralayer scattering caused by the graphene layers themselves. Assuming that a higher temperature results in growth and higher order in both, the graphene layers and in the layer stacking [20,37–43], overall, the reflections of samples get sharper with higher treatment temperature. For instance, PF-R 1000 showed broad reflections, while PF-R 3000 exhibits sharper reflections (Figure 5).

Additionally, the influence of the precursor can be studied for these two types of carbon: the PF-R-based carbon was prepared from phenol and formaldehyde, but the pitches consisted mainly of different aromatic molecules as building blocks (Figure 6). Therefore, the amount of foreign atoms, especially of hydrogen and oxygen, which prevents the formation of highly ordered graphene layers [37], was higher in the PF-R precursor than in the pitches. Moreover, PF-R polymerized in a first step without building $sp^2$-hybridized layers. By contrast, in pitch-based carbons, the dimensions and order of the graphene stacks increase faster with increasing heat treatment temperature, as is qualitatively seen in Figure 2B,C. In the following chapters, we address the question as to whether structural parameters and the evolution of the graphene structure can be correlated with differences in the content of foreign atoms, assuming that a high amount of hydrogen hinders the formation of well-defined and highly ordered $sp^2$ layers. Evidently, the significant width of the reflections of the PF-R samples at higher *s*-values speaks to a relationship between the

impurities and the degree of disorder, especially for the lower temperatures of this series (1000 °C and 1200 °C).

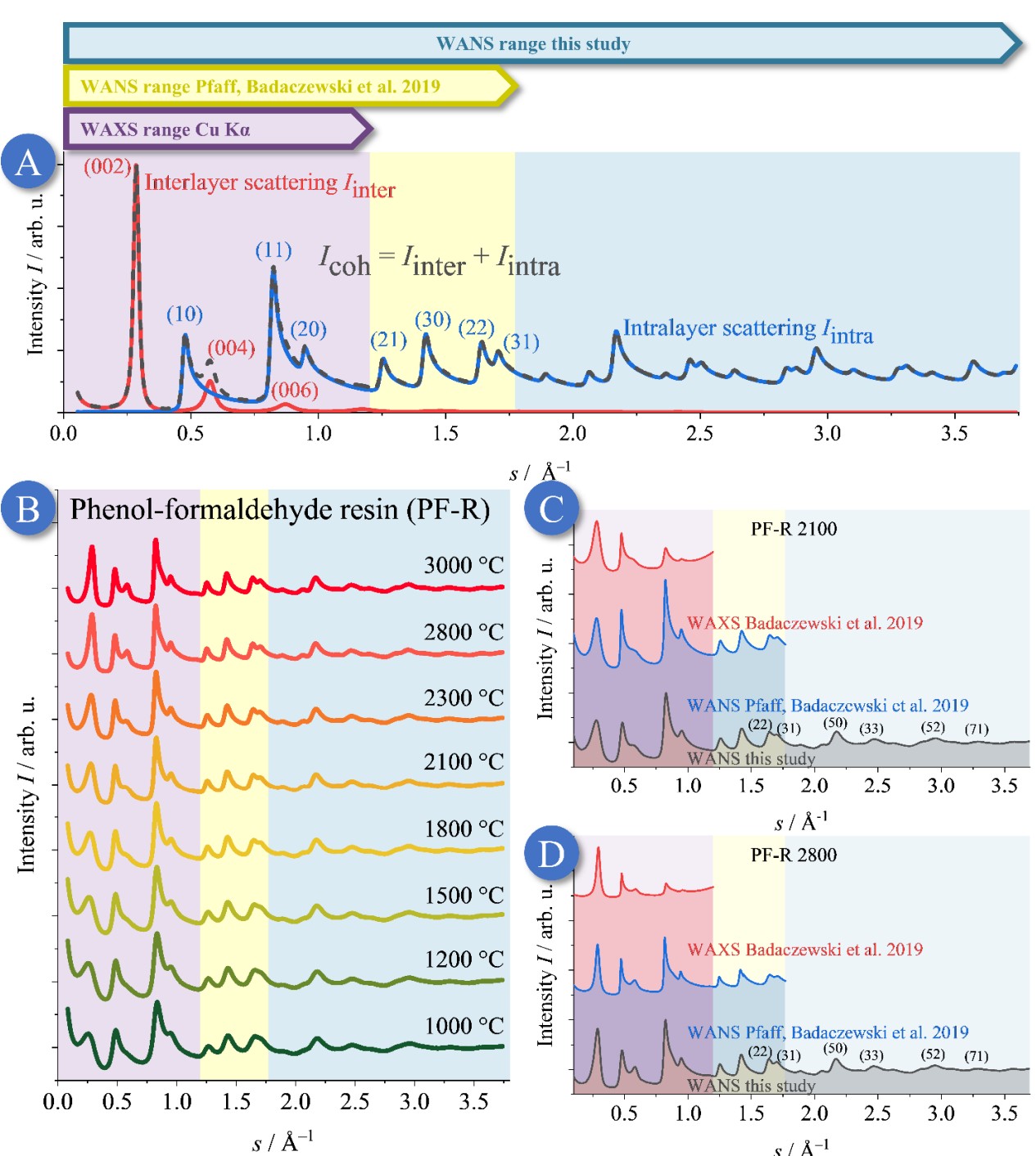

**Figure 5.** (**A**): The theoretical WANS intensity normalized per carbon (in electron units) $I_{e.u.}$ is given by a superposition of the interlayer ($I_{inter}$) and intralayer ($I_{intra}$) scattering. (**B**): WANS data of the phenol-formaldehyde resin (PF-R) carbon samples, including the previous WAXS (purple), the WANS (yellow) data, and the additional WANS range of the present study (blue). The temperature values represent the respective maximum heat treatment temperature. All these WANS data were already corrected for incoherent scattering using the procedure shown in Section S1 in the SI. (**C,D**): Comparison between the previously published WAXS (blue) and WANS (red) data from Pfaff et al. (2019) (blue), and the WANS measurements of this study, acquired at the Grenoble facility (black).

Since in this present study, the accessible *s*-range was much larger than in Pfaff et al. (2019) [43] and Badaczewski et al. (2019) [37], the intralayer structure could be determined more accurately. By contrast, for the stacking, no improvements were possible, as the widths of the (00*l*) reflections for these carbons increased so strongly with increasing *s* that these reflections were dampened out at moderate *s*-values. The WANS data for the temperature series of the LSPP and MP samples (pitches) can be found in the SI file (Figures S6 and S8).

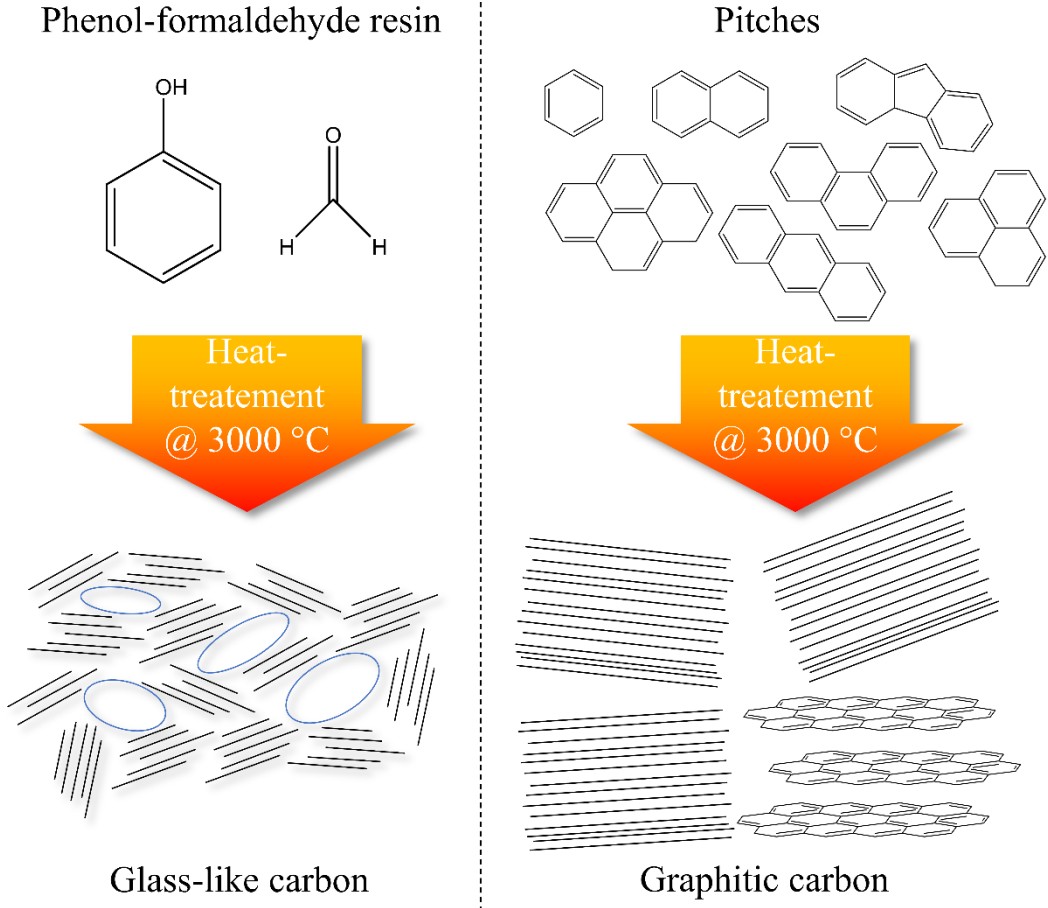

**Figure 6.** Schematic representation of the two types of precursors used. Phenol-formaldehyde resins form slightly porous glass-like carbons at high temperatures, while pitches, especially low softening-point pitches, can form graphite at very high temperatures.

The main advancement of this study using a beamline at the Grenoble neutron facility, when compared to our previous WANS measurements at HZB (Berlin), is the acquisition of reflections of up to quite high *s*-values, where mainly (*hk*) reflections contribute to the WANS pattern. Therefore, more reflections were used, allowing accurately determining the layer extent ($L_a$) and disorder ($\sigma_1$) from the (*hk*) reflections, using the Grenoble WANS data. Nevertheless, these WANS data suffered from a lower resolution for lower *s*-values. Since, however, the low-*s*-range is necessary to precisely determine the interlayer stacking, the evaluation of the WANS data from Pfaff et al. (2019) [43] was assumed to be more accurate at small *s* values, i.e., for determining the interlayer parameters ($L_c$, $\sigma_3$). Therefore, to achieve a maximum precision in quantifying the interlayer as well as the intralayer parameters, we combined the data of the two WANS measurements acquired from identical samples, the procedure of which is described in the SI file (S3). It should be noted that the indexing shown with parentheses strictly denotes lattice planes, and reflections would have to be indexed without parentheses. However, brackets are used in this publication for both meanings to improve readability.

*3.2. Non-Graphitic Carbons from Phenol-Formaldehyde Resin (PF-R)*

Here, the results for the phenol-formaldehyde resin (PF-R) temperature series will be discussed (Figure 7). Overall, the stack height ($L_c$) and layer extension ($L_a$) increased with increasing heat treatment temperature, while the disorder parameters in the stacking and layers ($\sigma_3$ and $\sigma_1$, respectively) decreased. Additionally, the average layer distance ($\overline{a_3}$) decreased, while the minimal layer distance ($a_{3\,min}$) increased as well. These changes were in conformity with the assumption that the degree of order, as well as the dimension of the graphene stacks, raises with increasing temperatures. PF-R 1000 contained a significant amount of hydrogen, which generated a substantial background scattering in WANS and required suitable correction. We found that this procedure rendered the determination of the average layer distance $\overline{a_3}$ to be uncertain, but all other parameters did not seem to be influenced by the high hydrogen content. In the following, the parameters and trends upon temperature treatment are discussed and compared to our previous WANS/WAXS series for these samples.

3.2.1. Graphene Stacks of the PF-R Carbons Temperature Series

Notably, our previous and current WAXS/WANS studies (Figure 7) yielded similar trends in the structural parameters as a function of treatment temperature, even with the absolute values being different (beside PF-R 1000, as described above). While the $L_c$ values were similar, this was not the case for the stacking disorder ($\sigma_3$). Note that a higher value of $\sigma_3$ indicates a higher degree of disorder. As found by Badaczewski et al. [37], the parameter value decreased from ~0.5 to 0.15 Å, whereas, in this study, the value decreases from 1 (upper limit) to ~0.8 Å. These different absolute values thus suggested a higher degree of disorder, but such straightforward interpretation ignores another fitting parameter, which is the minimal layer distance ($a_{3\,min}$), which had been reported in previous studies of such carbon materials [20,37,39–41,43]. Based on the excellent data quality in this present study, we found that the dimension of refined $a_{3\,min}$ values was reasonable, in the range of the ideal layer distance in graphite. However, the uncertainty in $a_{3\,min}$ as obtained from the refinement was so high that no convincing relationship to the treatment could be deduced. Additionally, the average layer distance ($\overline{a_3}$) is directly linked to the position of the reflection using Bragg's law and should, therefore, not change [57]. Interestingly, the $L_c$ value slightly decreased in the temperature range of 500 °C 1000 °C, and then continuously increased. This decrease can be attributed to the release of pyrolysis gases in this temperature range, which results in a certain disruption of the graphene stacks, as has been reported previously [37].

The uncertainty for $\overline{a_3}$ as a refined parameter was between 5% and 10%, which decreased with higher heat treatment temperature. This tendency can be explained by the interplay of ($a_{3\,min}$) and the standard deviation of the layer distance ($\sigma_3$). Regarding to the work of Ruland and Smarsly [34], all these parameters contribute to the profile shape of the (00$l$) reflections, and $a_{3\,min}$ has an especially significant influence on the peak position. Therefore, the refined values for $a_{3\,min}$ and $\sigma_3$ in this study might be different compared to the references. However, the values of $\sigma_3$ possess a lower uncertainty and can thus be used to compare the samples with respect to this type of translational disorder.

Details about the analysis of the polydispersity of the stacks ($\kappa_c$) and the homogeneity ($\eta$) can be found in the SI file in S1.

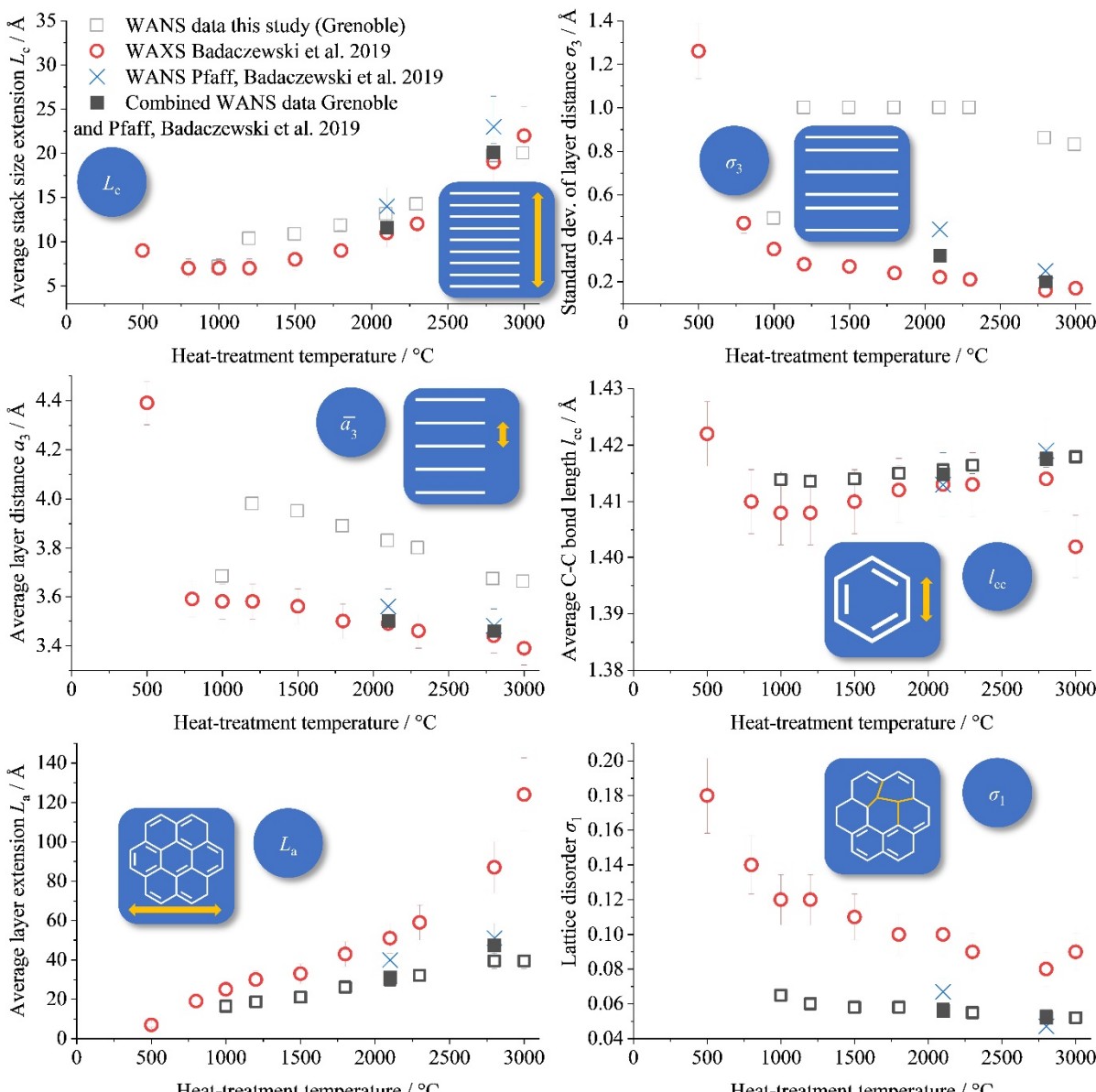

**Figure 7.** Microstructural parameters obtained from WANS refinement for the carbons derived from phenol-formaldehyde resin (PF-R) as precursor (black border only: this study (WANS Grenoble), red: Badaczewski et al. (WAXS) [37], blue: Pfaff et al. (WANS Berlin) [43], black filled: combined WANS data from Grenoble and Berlin). More information about the meaning of the absolute value of $\sigma_1$ can be found in the SI file in S6. The decrease in $L_c$ for temperatures between 500 °C and 1000 °C might have been caused by the release of pyrolysis gases in this temperature range, as already described previously [37]. This interpretation is supported by the quite high values of $\sigma_3$ for 500 °C.

### 3.2.2. Layer Structure of the Phenol-Formaldehyde Resin (PF-R) Temperature Series

For the microstructural parameters specifying the graphene layers, similar conclusions can be made as for the stacking with regard to the comparison with our previous studies. While the average C-C bond length ($l_{cc}$) was nearly identical to the results of Badaczewski et al. [37], the values for the layer disorder ($\sigma_1$) and the average layer extension ($L_a$), obtained in this study, were significantly lower. Additionally, the polydispersity of the layer extension ($\kappa_a = 1/\nu$) was smaller, since $\nu$ was set to seven to follow the work of Osswald and Smarsly [21], and, therefore, $\kappa_a$ was systematically smaller. However, the influence of $\kappa_a$ on the WAXS/WANS data is only small; it is rather $L_a$ and $\sigma_1$ that determine the (hk) profiles.

Smaller layer dimensions ($L_a$) lead to broader reflections; a smaller value of $\sigma_1$ causes the opposite, i.e., sharper (hk) reflections. As a difference, the layer dimension $L_a$ influenced all (hk) reflections identically with respect to the width, while—for fundamental reasons - the impact of $\sigma_1$ on reflections width continuously increases at higher s values [34]. Since the measured s-range was much higher for our WANS data than for the WAXS data, and due to the absence of the damping of the atomic form factor, the intralayer parameters for the layer structure were determined much more reliably in this study than in the reference studies.

To illustrate the influence of the experimental data range, the measured and refined data for the PF-R 3000 are shown as a representative sample (Figure 8). Additionally, simulated WANS data using the microstructural parameters from Badaczewski et al. [37] (refined from WAXS data) are included. Hence, beyond the accessible data range, these simulated data are a projection calculated with the refined set of parameters based on WAXS data featuring a shorter s-range. It can be seen that the higher order (hk) reflections were broader compared to the actually measured WANS reflections. This deviation can be explained by the parameter $\sigma_1$, which has a higher influence at higher order reflections at a large s. The refinement of the WANS data reveals that $\sigma_1$ was substantially smaller than determined in our previous study, and, concomitantly, $L_a$ was significantly smaller, too, which is further illustrated in Figure 9A. In detail, the $sp^2$-hybridized graphene sheets had markedly smaller extensions ($L_a$~16–40 Å for 1000 °C–3000 °C, compared to 7–87 Å), but possessed a higher degree of order ($\sigma_1$~0.065–0.05, compared to 0.18–0.08) compared to Badaczewski et al. [37]. An interpretation of different values for $\sigma_1$ is discussed in the SI in S6.

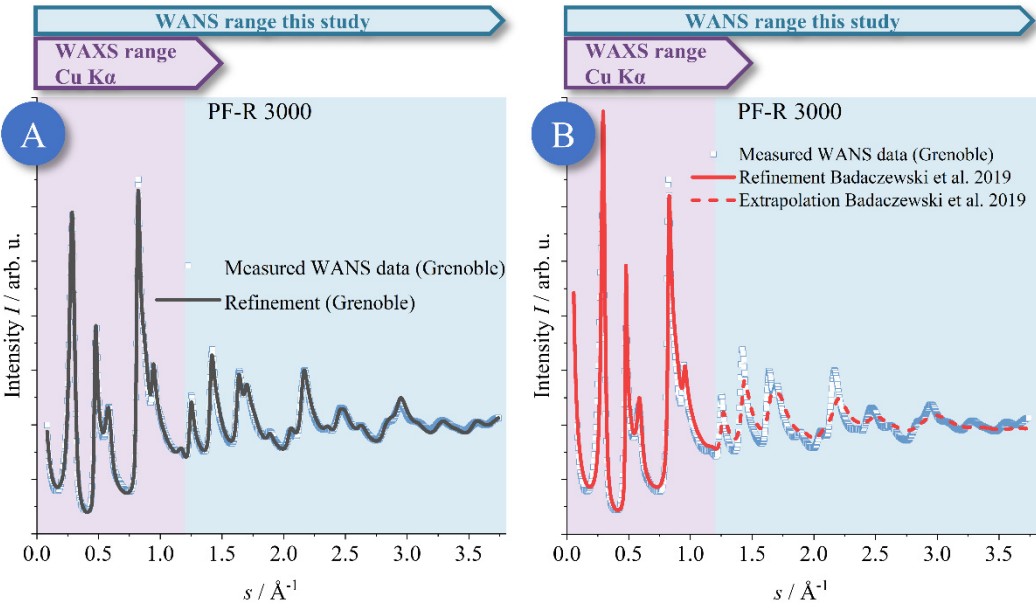

**Figure 8.** (**A**): Measured and refined WANS data of PF-R 3000. (**B**): Measured WANS data with simulated WANS data calculated from the WAXS refinement of Badaczewski et al. [37]. The dotted curve means an extrapolation in the range of large s, which was not measured in the reference.

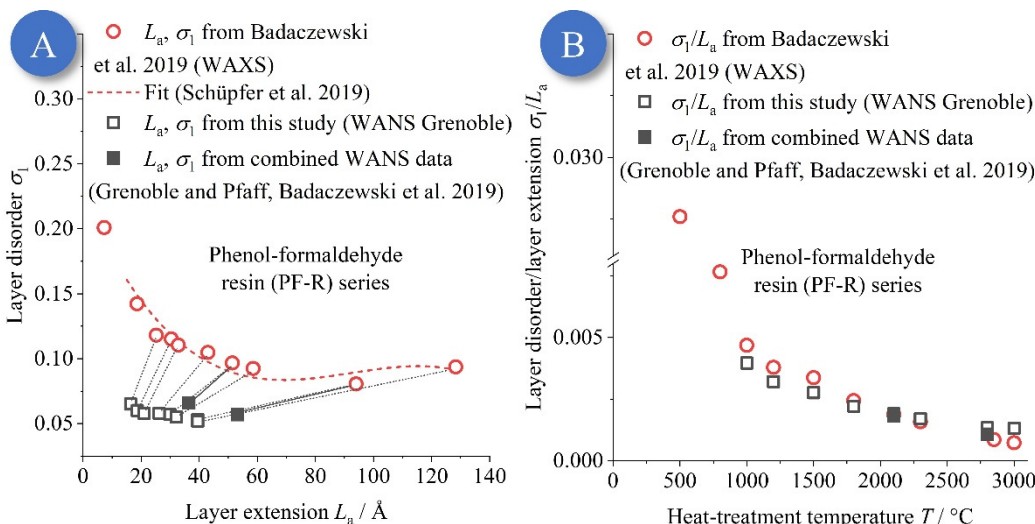

**Figure 9.** (**A**): Plot of $\sigma_1$ vs. $L_a$ for the phenol-formaldehyde resin (PF-R) temperature series. determined by WANS and previous WAXS results from Pfaff et al. 2019 [43] and fitted by Schüpfer et al. 2019 [25]. This comparison illustrates that the $\sigma_1$ values were much smaller than previously assumed. (**B**): However, the ratio $\sigma_1/L_a$ did not change, compared to our previous studies, over the whole temperature range. This agreement is due to the fact that it is these two parameters which determine the profile of the (*hk*) reflections, i.e., their widths, especially at large *s* values. A similar figure for the pitches can be found in the SI file as Figure S15.

Figure 8 thus demonstrates that the range of *s* values exerts a dramatic impact on the evaluation of WANS/WAXS data with regard to the extraction of structural parameters. While the qualitative progression of layer extent and disorder as a function of treatment temperature is correctly described by our previous studies on the same samples (Badaczewski et al. [37]), the cut-off at moderate *s* values resulted in erroneous values for $L_a$ and $\sigma_1$. In other words, if the already-published parameters $L_a$ and $\sigma_1$ described were correct, the simulated data in Figure 8 should fit the measured WANS data over the whole *s*-range and not only for small *s* values. We conclude that the intralayer parameters determined from the present WANS data are significantly more accurate and reliable than our previous analyses based on WAXS.

Figure 9B shows the quotient $\sigma_1/L_a$, which depends on the heat treatment temperature, for the different analyses. This quotient does not significantly alter for the whole temperature range, which is reasonable and not surprising, because the width of the (*hk*) reflections is mainly determined by $L_a$ and $\sigma_1$. Also, the combination of the different WANS data did not influence the intralayer parameters significantly, since the experimental broadening in the WANS data from Grenoble was smaller for higher *s* values. The higher (*hk*) reflections did not suffer from this problem, i.e. $L_a$ and $\sigma_1$ could be determined accurately.

### 3.3. NGCs from Mesophase (MP) and Low Softening-Point Pitch (LSPP)

In general, pitch-based carbon (MP and LSPP) possesses a higher order in the graphene microstructure than the resin-based carbons (PF-R) at the same heat treatment temperature. Additionally, the low softening-point pitch (LSPP) exhibits a more ordered graphene structure than the mesophase pitch. This general behavior is already well known and illustrated in Figure 6. The phenol-formaldehyde resin contains phenol and formaldehyde, which polymerizes under the release of water. Nevertheless, the resulting carbon structure still contains hydrogen and oxygen (see elemental analysis in Section 3.5 or Table S3), especially at lower temperatures. The pitches mainly consist of aromatic molecules, which can build $sp^2$-hybridized graphene layers more easily than resin. Therefore, the layer extension is much higher for the pitches, and, additionally, at very high heat treatment temperatures, a graphitic carbon is formed, i.e., one that shows three-dimensional order.

Since the softening point of LSPP is much smaller than for the mesophase pitch (70 °C in contrast to 250 °C [40]) due to generally smaller polyaromatic molecules, the LSPP forms bigger graphene layers and stacks than MP. Contrary to PF-R-based carbon, the LSPP samples start forming graphitic carbon above temperatures of 2500 °C (see Figure S8), as is evidenced by the occurrence of reflections of the (*hkl*) type, i.e., a three-dimensional periodicity. Hence, for these samples, the model of Ruland and Smarsly [34] cannot be used. Thus, the LSPP 2800 and 3000 should be analyzed by a single-peak analysis. The LSPP 1800 and 2500 were borderline cases for both evaluation methods. Here, the LSPP 2500, LSPP 2800, and LSPP 3000 were included into the comparison of structural parameters (Figure 10) using a single-peak analysis, as they are not NGCs but graphitic carbons. This discussion should be kept in mind when dealing with the resulting absolute values.

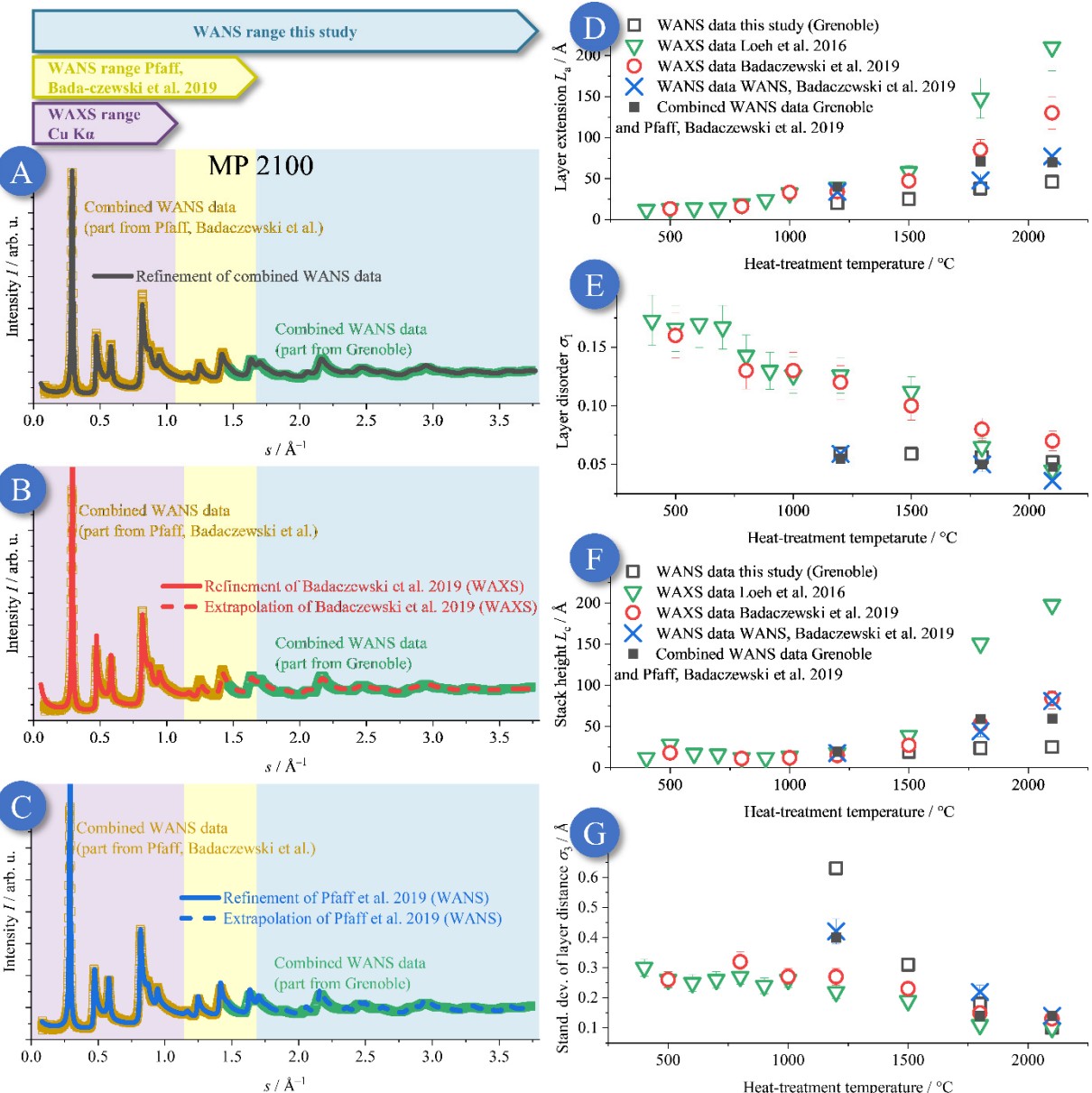

**Figure 10.** (**A–C**): Comparison of the mesophase pitch temperature series between the present WANS study, WAXS analysis of Badaczewski et al. [37], and WANS data from Pfaff et al. [43]. There are significant differences between the fits of this study (black border only) and the simulation using the WAXS results from Loeh et al. [40] (2016), Badaczewski et al. [37] (2019) and the WANS results from Pfaff

et al. [43] (2019). Especially for more highly ordered samples, i.e., higher heat treatment temperatures, the resulting structural parameters are quite different. These differences are caused by the different experimental accessible *s*-ranges, as well as by different experimental broadening of the WANS measurements [43]. Therefore, the WANS data from the two different WANS facilities (Grenoble, HZB) were merged to optimize the accuracy in the parameter determination. The results from the reference studies were used to simulate a WANS curve in their measurement range (filled) and to extrapolate them beyond the range (dotted curve). It can be seen that the intralayer parameters could not be determined precisely using the prior measured data, since the extrapolation based on the previous studies did not fit the current WANS data taken at Grenoble facility. (**D**–**G**): Importantly, previous WAXS measurements from Loeh et al. and Badaczewski et al. [40] overestimated the layer size and disorder when compared to the present results determined from the WANS data from the beamtime in Grenoble. Due to the higher amount of visible (*hk*) reflections, it can be assumed that the results from these WANS data are more accurate. On the other hand, the experimental broadening in this study (Grenoble) at small *s* values led to too high and disordered stacks compared to WANS measurements from from Pfaff et al. [43]. A zoom of the WAXS region (purple) can be found in the SI file (Figure S15).

Comparison to Previous WANS and WAXS Studies

The works of Pfaff et al. [43], Badaczewski et al. [37], Loeh et al. [40] had analyzed the very same mesophase pitch samples by WAXS [37] and WANS [43], the results of which we now compared with the present WANS analyses (Figure 10). The resulting values can be found in the SI file in Figures S6 and S8, as well as in Table S1. Compared to the previous WAXS analyses, the values for the average layer distance ($\overline{a_3}$) and C-C bond length ($l_{cc}$) were comparable. However, for most of the other important structural parameters ($L_c$, $L_a$, $\sigma_1$, . . . ) there were marked differences, especially compared to the WAXS study of Badaczewski et al. [37], thus again proving the benefit of using WANS data for large *s* values.

To illustrate the importance of a large *s*-range as shown in in Figure 10, we compared the refinement of the combined WANS data (Grenoble, HZB, Figure 10A) with simulated WANS data (Figure 10B,C) using the structural parameters taken from the previous WAXS and WANS measurements from Loeh et al., Badaczewski et al., and Pfaff et al. [37,40,43] for the MP 2100. Thus, in the case of Figure 10B,C, the curves are an extrapolation beyond the respective maximum accessible value of *s* for the respective instrument. These reflections were too broad in case of the extrapolation using the parameters from Badaczewski et al. [37] as determined by WAXS (Figure 10B). In contrast, for the simulation using the WANS results from Pfaff et al. [43] (Figure 10C), the simulated reflections at higher *s*-values were too sharp. In conclusion, too big and ordered layers were obtained in the WANS study of Pfaff et al. [43], and too disordered layers were proposed in the WAXS study of Badaczewski et al. [37], as shown in Figure 10D,E. Magnified figures further illustrating such comparisons are given in Figures S14 and S15.

Overall, it can be concluded that the presence of higher-ordered (*hk*) reflections is crucial to determine the intralayer parameters exactly, but a high *s*-space resolution over the entire *s*-range is crucial to determine the interlayer parameters exactly. Hence, ideally, a valid determination of all microstructural parameters can only be reached by using a combination of two different WANS measurements, for the small and large *s*-range, as described in Section 2.2.

Since WAXS study of Loeh et al. [40] published only results for lower heat treatment temperatures of LSPP, only a few LSPP samples could be compared with the results of the present WANS study (Figure 11). However, a detailed comparison was possible with the WAXS analysis from 2019 [37] and the WANS analysis of Pfaff et al. [43]. The differences for the microstructure data of the low softening-point pitch (LSPP) at a heat treatment temperature of 1200 °C (LSPP 1200) were only small (Figure 11). In contrast to this, the results for LSPP 1800 were very different. These differences can be explained by the

experimental broadening from the WANS data collected in Grenoble. The combination with the WANS data from Berlin lead to more reliable and accurate results for the intralayer parameters.

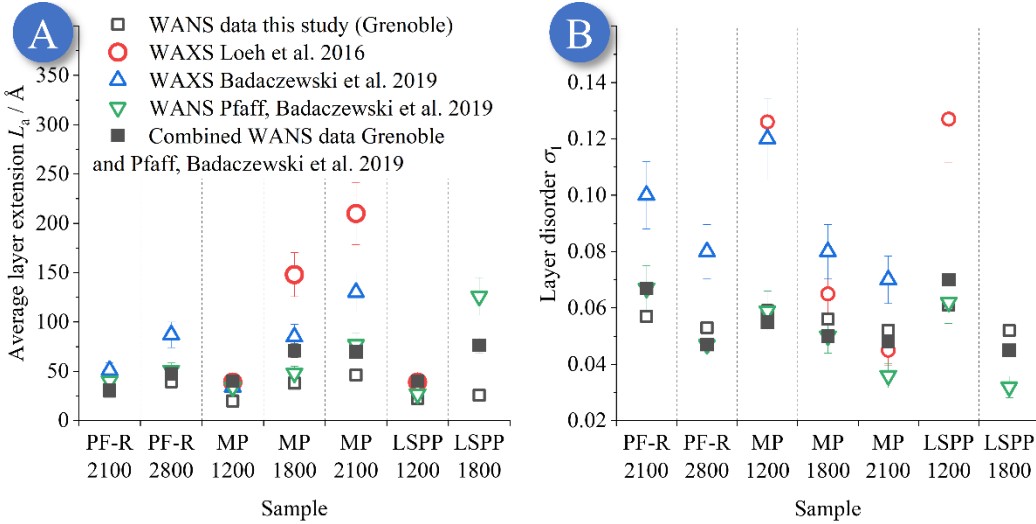

**Figure 11.** (**A**,**B**): Comparison of $L_a$ (**A**) and $\sigma_1$ (**B**) values obtained from different scattering experiments using X-ray and neutron radiation for identical samples. Black border only: WANS results from WANS data collected in Grenoble. Red: WAXS results from Loeh et al. (2016) [40]. Blue: WAXS results from Badaczewski et al. (2019) [37]. Green: WANS results of Pfaff et al. (2019) [43]. Black filled: results from the combination of WANS data from Grenoble and Pfaff et al. [43]. In general, WANS measurements lead to more accurate results for the layer structure because of the larger accessible $s$-range and the absence of the atomic form factor.

For LSPP 2500/2800/3000, we only used the average stacking height $L_c$ as determined by Scherrer analysis from the (002) reflection (Equation (2)) for comparing the samples (Figures S7 and S8, Table S1). As for the other samples and parameters, the overall tendency, i.e., an increase in $L_c$ following an increase in heat treatment temperature, was identical to the previous studies, but the absolute values were different. These differences might have been caused by the different approaches according to the fit function (Gauss vs. Voigt) and the different resolution $\Delta s/s$, which both influenced the resulting parameters. However, the value for $L_c$ determined from the WANS data from Grenoble seems to be too high for the LSPP 2500, which was caused by the degree of disorder in this sample. At this heat treatment temperature, no three-dimensional crystallographic order was present, and the resulting stack height $L_c$ was intrinsically overestimated by the Scherrer equation. Additionally, the average layer distance ($\overline{a_3}$) showed a clear decreasing tendency down to 3.38 Å for the LSPP 3000, which is close to the theoretical value of perfect graphite (3.35–3.36 Å) and corresponds to a graphite-like microstructure [58,59].

*3.4. Structural Differences between the Resin and the Pitches*

Overall, there was a clear tendency of increasing domain sizes and decreasing disorder from the phenol-formaldehyde resin (PF-R) over the mesophase pitch (MP) to the low softening-point pitch (LSPP) carbons. For the interlayer structure, the parameters $L_a$ (average layer extension) and $\sigma_1$ (layer disorder) are most relevant. For a treatment at 2100 °C, $L_a$ increased from 16 to 40 Å for the PF-R in contrast to 20 to 45 Å for the MP, and even higher values that were found for the LSPP. Owing to the occurrence of general (*hkl*) reflections for the LSPP 2500/2800/3000, but not for the PF-R samples treated at such temperatures, $L_a$ could not be calculated and compared for these samples in a consistent way. The observed differences were caused by the different precursors, as pitches contain a higher proportion of aromatic molecules in the precursor with a relatively low concentration

of foreign atoms. Thus, these polyaromatic hydrocarbons grow faster to extended $sp^2$-hybridized layers. For both the PF-R and MP, $\sigma_1$ significantly decreased upon treatment at 2100 °C, from $\sigma_1 = 0.065$ to 0.052 (PF-R) and $\sigma_1 = 0.059$ to 0.0482 (MP), which indicates higher-ordered graphenes.

In contrast to the interlayer structure, the parameters $L_c$ (average stack height) and $\sigma_3$ (stacking disorder) are the most significant ones regarding the interlayer structure. Surprisingly, and contrary to expectation, our WANS analysis revealed that, for the MP and LSPP carbons, $L_c$ remained quite small until reaching considerably high temperatures (see Figure 10, Figures S7 and S8), did not exceed ~60 Å at up to 2200 °C, and only increased up to ~70 Å for a heat treatment temperature of 3000 °C (LSPP). Yet, the $L_c$ values for the MP/LSPP were significantly larger than for the PF-R carbons and rose moderately from 7 to 20 Å (3000 °C). Interestingly, the disorder of the stacks was still higher for the MP samples than for the LSPP carbons, which can be explained by the different softening points (250 °C for MP vs. 70 °C for LSPP). In general, the softening point influences the carbonization and pyrolysis at lower temperatures, as a lower thermal energy is needed to build $sp^2$-hybridized layers and small stacks, as is the case for materials with lower softening points [40]. In contrast to the pitches, PF-R-derived carbon contains more foreign atoms and less aromatic hybridized molecules, which is related to the larger degree of disorder and smaller stack dimensions. See the following section for more details. Additionally, these materials can contain nanoscaled porosity, implying that the resulting density is significantly lower than for the pitches or graphite [37].

### 3.5. Comparison of Structural and Chemical Analysis

Elemental analysis made it possible to determine the proportions (mass) of hydrogen, oxygen, nitrogen, and sulfur (the latter only for the LSPP). Overall, the amount of foreign atoms decreased with an increasing heat treatment temperature (Figure 12). In the case of the LSPP, sulfur was additionally detectable, which is caused by the precursor, but the amount was so small that it will not be considered in the further analysis. The elemental analysis data for the PF-R were already published by Badaczewski et al. [37], and those for the LSPP were only published for lower temperatures [40]. The other values were derived in the present study. The results for all samples are shown in Figure 8, Figures S1–S3, and Table S3. It should be emphasized that we focused on temperatures above 1000 °C, which pertain to quite low contents of foreign atoms below 1 wt.%. Still, even such small fractions seemed to exert a strong impact on the formation of the graphene layers, as is discussed in the following. Notably, a small amount of only 0.5 wt.% hydrogen impeded the analysis of WANS data due to the huge incoherent background.

As a first conclusion for the PF-R samples, there is a clear correlation between the graphene sizes/disorder and the concentration of hydrogen/oxygen/nitrogen (Figure 12): along a decreasing content of these impurities, the stacks/layers continuously become bigger and more ordered. Note that the hydrogen and oxygen fractions are not independent, as hydrogen is linked to oxygen-containing moieties such as hydroxyl or carbonyl groups, but it is also bound to carbon at the edges of the graphene sheets. Hence, the lateral growth of graphenes should go hand in hand with the removal of hydrogen.

In detail, the $L_a$ values for the PF-R (Figure 12) were still quite small for a treatment at 1000 °C, being on the order of 1–2 nm only, as was already found in previous studies (Loeh et al., Badaczewski et al., Pfaff et al. [37,40,43]). At higher temperatures, the substantial growth of the layers, as well as a decrease in the disorder parameter $\sigma_1$, were observed, which corresponded to a decline in the fraction of hydrogen/oxygen below 0.5 wt.%. Our previous studies suggested that hydrogen and oxygen-containing molecules such as $CO_2$ and $H_2O$ are released at temperatures up to 1000 °C [37].

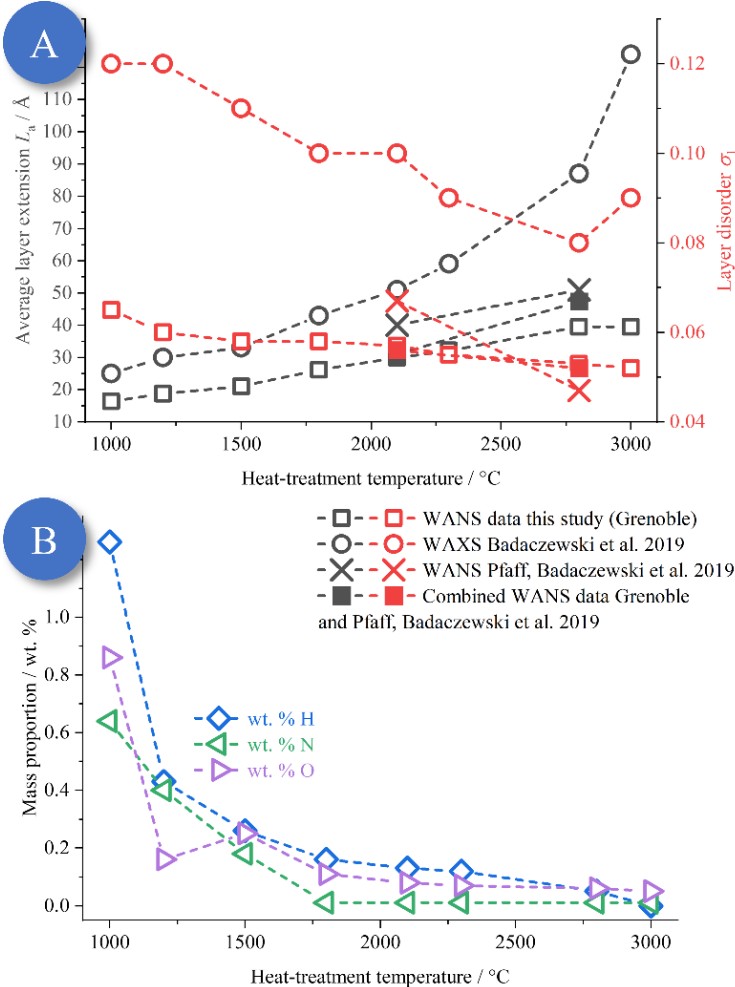

**Figure 12.** (**A**): Comparison of the layer extension ($L_a$) and disorder ($\sigma_1$) with elemental analysis (**B**) for phenol-formaldehyde resin (PF-R) as precursor. Foreign atoms such as oxygen and nitrogen hinder the formation and lateral growth of graphenes (in the form of carbonyl groups, -OH groups, etc.). Therefore, a decreasing concentration of elements leads to bigger and more highly ordered graphene layers. The values of $L_c$ and $\sigma_3$ are not shown here, because the trends were quite similar to ($L_a$) and disorder ($\sigma_1$). The corresponding figures for MP and LSPP can be found in the SI file (Figures S12 and S13). The data used can be found in the works of Badaczewski et al. (2019) [37] and Pfaff et al. (2019) [43].

For both pitches (Figures S12 and S13), similar trends were observed; thus, only a small number of samples were studied. For these pitch samples, the amount of foreign atoms decreased upon higher heat treatment temperatures, and the absolute amount was significantly smaller than for the PF-R samples at the same temperatures. This smaller content of foreign atoms was accompanied by larger stack/layer dimensions and higher order when compared to the resin-based carbon samples. Interestingly, nitrogen was still present at higher heat treatment temperatures for all samples, and a small residual amount remained, even at high $T$. In addition, only for the LSPP samples at 2500 °C and above, "mixed" (*hkl*) reflections occurred, which speaks to a graphitic structure. At this point, hydrogen was no longer detectable. For the pitches, a further parameter needed to be considered in the evolution of the microstructure next to the content in foreign atoms, which was the softening-point itself, as seen by the differences between MP and LSPP.

In contrast to this, the PF-R as a precursor led, in general, to glass-like carbon through to the pyrolysis process at lower temperatures. This process is caused by the significant amount of oxygen/nitrogen in the precursor even at higher temperature, as well as by

the lower proportion of aromatic/$sp^2$0hybridized species, thus hindering the formation of graphene layers [37]. Closer insight was obtained by comparing the structural parameters $L_a$ and $\sigma_1$ with the composition (Figure 12): while $\sigma_1$ stayed almost constant up to high temperatures, $L_a$ continuously rose, while the amount of foreign atoms declined slowly in the range of 2000 °C. Thus, the changes in composition seem primarily be related to the lateral growth of the graphenes, while, surprisingly, their order ($\sigma_1$) was already quite developed and high as for the pitch-based carbons (Figure 11) but was not further enhanced at very high treatment temperatures (above 2500 °C). Hence, the thermal energy seems to be decisive for removing foreign atoms from the graphenes' edges and for the conversion of $sp^3$ into $sp^2$ carbon.

Overall, based on the comparison of the elemental analysis of the determined microstructure data, the following can be concluded:

1. The stack height, stack disorder and the layer extension/disorder are strongly linked to the amount of foreign atoms, especially oxygen. Due to the very low amount of foreign atoms at this temperature, $sp^3$-hybridized domains probably cause this disorder.

2. A detailed comparison of composition and disorder, as described by the parameter $\sigma_1$, is only meaningful using valid microstructural data, which could not be obtained in previous studies, except for by using the high-quality WANS data in this study.

3. Oxygen or oxygen-containing functional groups are located on the edges as well as on the lower/upper side of the graphene layers; otherwise, either the layers or the stacks would have to grow faster at high temperatures, especially for the PF-R carbon.

4. Oxygen is believed to often be removed as water or other functional groups that contain hydrogen. While hydrogen is still required at the edges of the graphene layers to saturate the free electrons, the hydrogen content nonetheless decreases significantly. Therefore, oxygen and hydrogen are believed to be removed together. Similar trends for lower heat treatment temperatures have been shown and proven in previous studies [37,40].

5. Nitrogen has no direct correlation with the stack/layer size and order. Therefore, nitrogen must be mainly built in into the graphene structure. This is possible and plausible, since nitrogen can also make $sp^2$-hybridized structures.

6. For the formation of three-dimensional ordered graphite, the amount of foreign atoms must be small, close to zero. It seems that the absence of hydrogen (or a not-detectable amount) is a good indicator.

7. At very high temperatures (2500 °C and above for LSPP), it is not the foreign atoms, but the heat treatment temperature, i.e., the thermal energy, that causes the higher order of the graphite stacks.

### 3.6. Raman Spectroscopy

Raman spectroscopy analysis was performed using the procedure that was recently introduced by Schüpfer et al. in 2020 [25]. In that study, several of the samples had already been investigated, and an improved methodology was established for the interpretation of Raman spectroscopy data in terms of $L_a$ and $\sigma_1$ by comparison with the WAXS analysis. Hence, here we used the advanced precision in the determination of the $L_a$ and $\sigma_1$ by WANS analysis to check and validate the procedure of Schüpfer et al. [25]. For the PF-R and LSPP, the already published data were used, but for the MP series, Raman spectroscopy data were measured in this study. Overall, the used spectral range of 1000 cm$^{-1}$–3000 cm$^{-1}$ shows the typical, most common D, G, D' and 2D bands (Figure 13 and Figure S18) [24,25,60–63]. As with most prominent signals, a broad and overlapping D (~1350 cm$^{-1}$) and G (~1850 cm$^{-1}$) band were observed for lower treatment temperatures for all samples. The absence of a 2D band up to this temperature indicated small and disordered graphene sheets, which matched the WAXS/WANS analysis [25,61,64]. For higher treatment temperatures, the D' and 2D bands appeared starting from ~1800 °C. At even higher temperatures, the signals became sharper, and, especially for the LSPP (Figure S18), the D and D' bands even

disappeared, which was due to a highly ordered microstructure in the pitch-based carbons, which coincided with the WANS analysis shown in the section above. A more detailed interpretation of the general trends for the temperature dependence of the Raman data of the PF-R and LSPP samples can be found in the work of Schüpfer et al., 2020/2021 [25].

In essence, Schüpfer et al. [25] correlated basic quantities of the Raman spectroscopic data of NGC, namely, the G- and D-band positions, the $I_D/I_G$ ratio, and the widths of the D- and G-band positions, against $L_a$ and $\sigma_1$. Owing to the large number of PF-R- and pitched-based carbon, a large range of $L_a$ values provided a meaningful basis for such correlations. This present study achieved an improved accuracy in determining $L_a$ and $\sigma_1$ and, thus, enabled a revision of the dependences established in ref. [25] when we recall that the WANS analysis on the Grenoble data yielded significantly smaller $L_a$ values for the very same samples. In the case of the $I_D/I_G$ ratio for the PF-R (Figure 13A), the general trend of an increase in $I_D/I_G$ ratio with increasing $L_a$ was confirmed. However, we observed a small, but significant shift of the overall dependence to smaller values of $L_a$ when compared to ref. [25], while, for the LSPP and MP samples (Figures S6 and S8 in the SI), the new values lay well within the proposed relationship. Our study thus supports the relationship between the $I_D/I_G$ ratio and $L_a$ for small $L_a < 6$ nm, but unfortunately for $L_a > 6$ nm, further samples need to be studied to determine the course of this relationship.

For the PF-R, the $I_D/I_G$ ratio (Figure 13A) increased up to a temperature of 1800 °C, and, additionally for the PF-R and LSPP, a clear blue shift was visible up to this temperature (Figure 13B). Additionally, the shape of the 2D signal was strongly symmetric and, therefore, a Lorentzian curve could be used to fit these data. The resulting full widths at half maximum (FWHM) were broader than in graphene, which indicates the turbostratic structure, thus independently confirming the WAXS and WANS analysis [25,34,65]. The asymmetrical shape of the 2D band for the LSPP at 3000 °C, where the right shoulder was higher than the left, is a clear indicator for a graphitized carbon structure [61,64].

Also, the measured D-band position depending on the average layer extension ($L_a$) was significant lower for the PF-R series when compared to the study of Schüpfer et al. [25], while for the LSPP, the measured values fit the theoretical values (Figure S18). However, for a highly graphitizable precursor (LSPP), the position of the D-band fit the theoretical values well, and the layer size could be determined from Raman measurements.

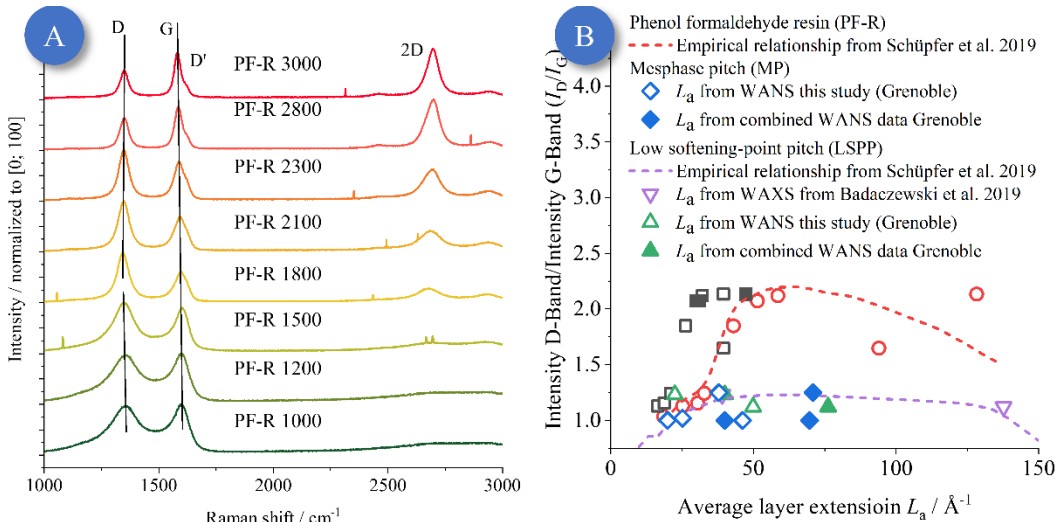

**Figure 13.** *Cont.*

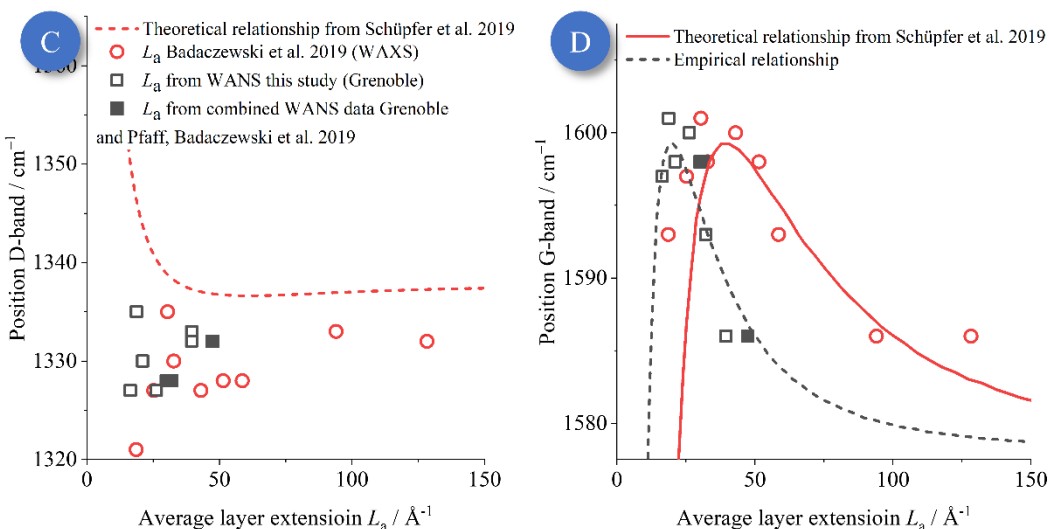

**Figure 13.** (**A**): Raman spectroscopy data for the phenol-formaldehyde resin-based carbons (PF-R) for different treatment temperatures, using an excitation wavelength of 633 nm. All samples showed increasing 2D and D' bands and a higher G/D ratio at higher heat treatment temperatures, which are clear indicators for the increase in layer extension and decrease in disorder at higher temperatures [24,25]. (**B**): The intensity quotients $I_D/I_G$ values were previously determined by Schüpfer et al. [25] and compared to the different $L_a$ values determined in this present study and in the work of Badaczewski et al. [37]. (**C**): The position of the D-band was lower than the theoretical values for these resins as determined in ref. [25], indicating that, for (disordered) glass-like carbon, such a theory may not be applicable. (**D**): Position of the G-band related to the $L_a$ values obtained from WAXS and WANS. Compared to the previous study of Schüpfer et al. [25], the G-band positions needed to be shifted to smaller $L_a$. In both cases, the Campbell–Fauchet approach was used to model the data, as described by Schüpfer et al. [25,66,67]. A similar figure for the pitches can be found in the SI as Figures S17 and S18.

For the G-band position, a similar shift of the entire relationship to smaller $L_a$ was observed. (Figure 13C). Hence, this significant deviation compared to the study of Schüpfer et al. [25] implies that the underlying Campbell–Fauchet approach [66,67] might need to be modified or might not even be applicable. From Figure 13C, one can extract a shift in the $L_a$ values by a factor of ca. two. Assuming that the Campbell–Fauchet approach is still valid, the constant in the exponent of the Fourier coefficient concomitantly would need to be changed by a factor of four, i.e., changing it formally to

$$C(\vec{k}_0, \vec{k}) \propto \exp\left(-\frac{1}{8}L_a^2(\vec{k}_0 - \vec{k})\right) \text{ to } C(\vec{k}_0, \vec{k}) \propto \exp\left(-\frac{1}{32}(0.5\,L_a)^2(\vec{k}_0 - \vec{k})\right).$$

Hence, the correction in the $L_a$ values shown in Figure 13D might point to an interesting aspect in the interpretation of band positions for graphene-like materials. One major intrinsic shortcoming of the Campbell–Fauchet approach, if applied to such materials, is the assumption of spherical particles, which is not valid for graphenes. Thus, our study might serve as the basis for a reconsideration of the important relationship between band positions and graphene layer dimension in general.

## 4. Discussion/Outlook: Usage of Small-Wavelength-Radiation for X-ray Scattering vs. WANS

In spite of the large *s*-range and high quality of the Grenoble WANS data, the question remains as to whether the refinement with respect to the most relevant intralayer parameters $L_a$ and $\sigma_1$ is accurate, or if an even lower wavelength is needed. Hence, we simulated WANS data, while varying $L_a$ and $\sigma_1$, for a carbon corresponding to the PF-R 1800 (Figure 14).

The refinement values for $L_a$ and $\sigma_1$ were used, and simulations for different values of $L_a$ and $\sigma_1$ were done, with the other parameters being kept constant (Figure 14A,B). Since $\sigma_1$ does not have a significant influence on the scattering data in the common WAXS range ($s < 1.2$ Å$^{-1}$), the layer disorder could be much overweighted or overestimated in WAXS measurements. For this reason, the average layer extension must be higher, otherwise the (10) and (11) reflections would not fit sufficiently. By using WANS data with a low radiation wavelength, a much higher range of $s$ is available, and, therefore, the intralayer parameters can be refined more exactly. To be more precise, the refinement is now unique, which was not the case when using only WAXS data. Moreover, it is now possible to disentangle $L_a$ and $\sigma_1$, i.e., the layer size and the layer disorder. Moreover, the Mo-K$\alpha$ radiation having a small wavelength and, therefore, allowing a higher measurement range cannot lead to more exact results, since the damping of the atomic form factor is too strong.

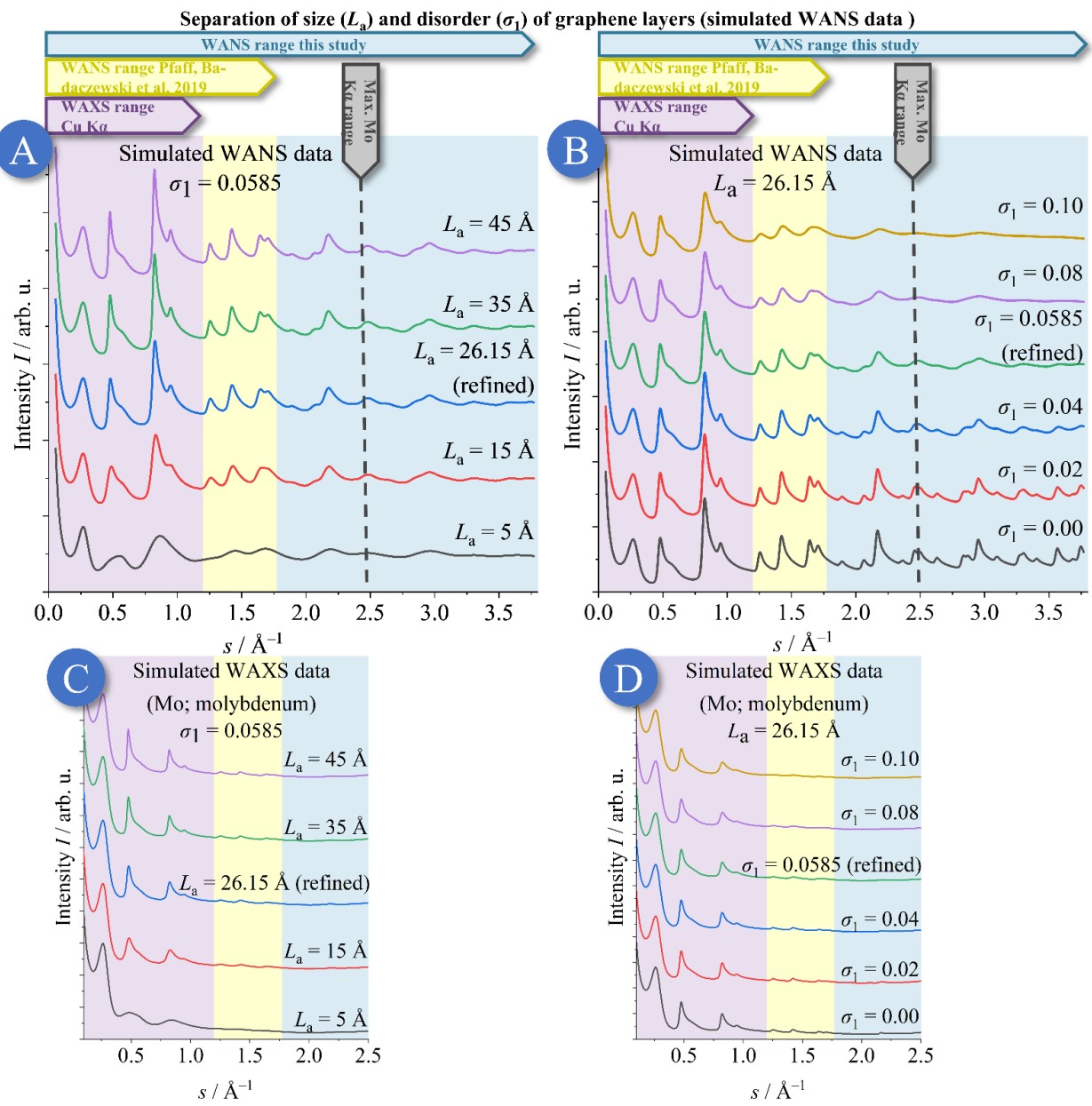

**Figure 14.** (**A–D**): Simulation of WANS and WAXS curves to assess the respective impact of different values of $L_a$ and $\sigma_1$ with regard to the fundamental question as to whether size ($L_a$) and

disorder/strain ($\sigma_1$) can be validly disentangled at all. The refined values for PF-R 1800 were used to simulate scattering curves with higher/lower layer extension/order. (**A**): simulated WANS data ($\lambda = 0.4975$ Å) with constant $\sigma_1$ and variation of $L_a$. (**B**): simulated WANS data ($\lambda = 0.4975$ Å) with constant $L_a$ and variation of $\sigma_1$ assuming typical values of $\sigma_1$. The layer disorder ($\sigma_1$) as well as the layer extension ($L_a$) were overestimated in previous WAXS studies, as the *s*-range was insufficient. By using WANS data presented here in this study, it is possible to differentiate between these parameters in their impact on the data and to quantify them reliably. Building on this leads to a new view of the NGCs, especially the resins: The graphene layers were much smaller but more ordered, as assumed before. (**C,D**): simulated WAXS curves using Mo-K$\alpha$ radiation ($\lambda = 0.71$ Å), under variation of $L_a$ and $\sigma_1$, analogue to (**A**) and (**B**). Using Mo-K$\alpha$ radiation ($\lambda = 0.71$ Å) allows for WAXS data with a high *s*-range, but suffers from the damping of the atomic form factor, so the visible reflections will have only a low intensity. Hence, size ($L_a$) and disorder ($\sigma_1$) cannot be precisely determined, even when using a Mo-K$\alpha$ radiation source. The measurement range of Pfaff et al. is related to reference [43].

In contrast to prior measurements and publications, these are the first measurements of such NGCs where the intralayer parameters could be determined validly and reproducibly. Since the layer disorder as well as the layer extension were overestimated in prior studies, the present analysis led to smaller, but higher-ordered graphene sheets.

## 5. Conclusions

This study was dedicated to the fundamental question of the magnitude of nanoscaled disorder in the abundant class of non-graphitic carbon (NGC) materials, which are composed of small-sized graphenes as building blocks. The basic structural make-up had already been described decades ago, based on X-ray scattering, by pioneers in the field, such as Rosalind Franklin and B. E. Warren: NGCs are made up of graphene stacks possessing finite, small dimensions, as well as rotational and translational disorder in the stacking of the graphenes. The substantial structural disorder of the graphenes themselves can be fundamentally quantified by analysis of the width of the (*hk*) reflections in wide-angle X-ray and neutron scattering (WAXS/WANS). However, as a finite dimension of the graphenes causes broadening of these reflections as well, the disentanglement of size and disorder from experimental WAXS/WANS data is a challenge.

The main strategy of this study was to merge WANS data from two facilities, HZB (Berlin) and ILL (Grenoble), that were acquired on the very same materials. Thus, the WANS data spanned a huge range of the modulus of the scattering vector *s* ($s = 2\sin(\theta)/\lambda$) and also profited from the different resolutions of the beamlines as small and large *s* values. We showed that WANS is superior to WAXS for several reasons, and that high-quality WANS data, obtained at the Grenoble facility, enabling a huge *s*-range, indeed allowed for disentangling size and disorder and, therefore, provided reliable values for the disorder in the form of the parameter $\sigma_1$. This led to a detailed and meaningful verification of the present view on the microstructure (especially the layer extension ($L_a$) and stack height ($L_c$), as well as their degree of disorder (mainly $\sigma_1$ and $\sigma_3$, respectively)) and their evolution at different heat treatment temperatures. We studied three different carbon materials from different carbon precursors, i.e., non-graphitizing glass-like carbon building (phenol-formaldehyde resin, PF-R) and two graphitizing pitch-based carbons (mesophase pitch and low softening-point pitch, MP and LSPP), which had been previously studied by WAXS and WANS [37,40,43].

Our study advances the characterization of graphene-based carbons and the view on their structural make-up in several aspects:

1.  Only WANS data with a high *s*-range lead to a reliable and reproducible determination of the dimension and disorder of the graphene layers. Using WAXS data or WANS data with a small *s*-range, e.g., using typical XRD instruments using Cu-K$\alpha$ radiation, leads e.g. to unreliable intralayer parameters $L_a$ and $\sigma_1$.

2.   The refinement approach of Ruland and Smarsly [34] allows for excellent fitting of WANS data, even up to quite large s values, which enables a precise determination of the microstructural parameters describing the graphene structure.

3.   Raman spectroscopy studies by Schüpfer et al. [24,25] established and advanced methods for the correlation between signal width/height of the different bands and the microstructure, especially $L_a$, based on a validation with WANS/WAXS. While the present study showed that the previously determined values for $L_a$ were too large, we demonstrated that the method by Schüpfer et al. is still usable to quantify the graphene structure and, in particular, to describe qualitative changes e.g., upon temperature treatment. Further theoretical work needs to be advanced in order to establish a quantitative relationship between band positions and the graphene dimension.

4.   As a most relevant finding, apparently "disordered" carbons (especially prepared from phenol-formaldehyde resin; PF-R) are much less disordered as previously assumed, based on the $\sigma_1$ values. Concomitantly, the graphene layers in such PF-R carbon samples are smaller (on the order of a few nm at most) but are internally more highly ordered than predicted in prior studies (Figure 15) in which the layer extension and disorder were overestimated systematically. We think this insight represents a paradigmatic, fundamental advance in the view on the structure of these materials. They are by no means "amorphous", but, on the contrary, the graphenes in NGCs possess an order close to an ideal "graphene". This result might contribute to the understanding of graphitization in the formation of graphite in the future.

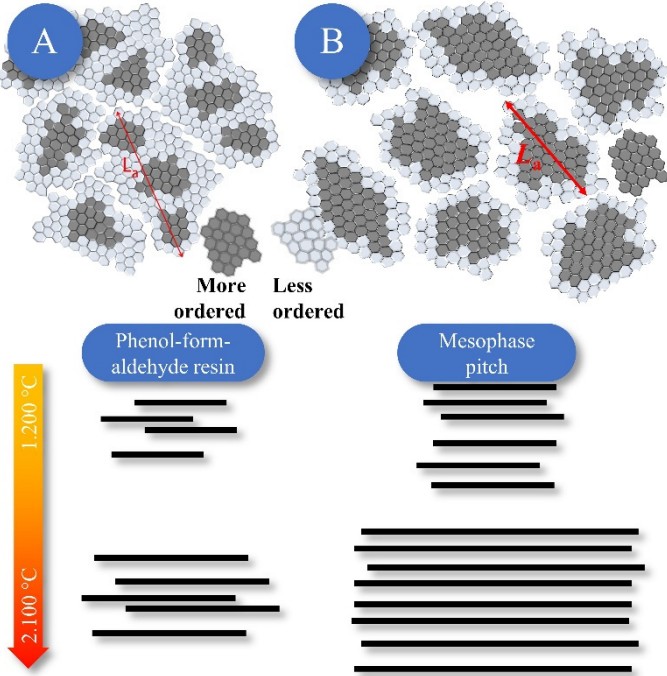

**Figure 15.** New insights into the microstructure of non-graphitic carbon: Compared to prior studies on the same materials [43] (**A**), the present analysis (**B**) suggests smaller, but highly ordered graphene-layers. Reprinted and adapted with permission from Ref. [43]. Copyright 2019 American Chemical Society.

Hence, our study stimulates further questions and further analysis e. For instance, the so-called pair distribution function (PDF) [49–53] will be calculated from the present WANS, but also from suitable WAXS data, for example, from an XPDF experiment performed at the Diamond light source. Additionally, WAXS measurements at similar wavelengths, as applied in this study (λ~0.5 Å), should be performed for samples containing a significant

amount of hydrogen, which impedes reliable WANS analysis. Tests should be performed as to whether the WAXS of such samples can be evaluated more accurately than at present and, additionally, if such measurements are as exact as WANS measurements.

In conclusion, we believe that the present study proposes an advanced methodology for studying graphene-based carbons, and also advances the view on the structure of important classes of non-graphitic carbon.

**Supplementary Materials:** The following supporting information can be downloaded at: https://www.mdpi.com/article/10.3390/c9010027/s1. SI—Additional Data and Mathematical Background. References [21,24,25,34,37,40,43,66,67] are cited in the supplementary mate-rials.

**Author Contributions:** Conceptualization, O.O. and B.M.S.; Data curation, H.E.F., A.F., J.-U.H. and M.R.; Formal analysis, O.O., M.O.L., F.M.B. and T.P.; Funding acquisition, P.J.K. and B.M.S.; Investigation, O.O., M.O.L., F.M.B. and T.P.; Methodology, O.O. and B.M.S.; Project administration, B.M.S.; Resources, P.J.K. and B.M.S.; Software, O.O.; Supervision, B.M.S.; Validation, O.O.; Visualization, O.O.; Writing—original draft, O.O.; Writing—review and editing, M.O.L., F.M.B., T.P., H.E.F., A.F., J.-U.H., M.R., P.J.K. and B.M.S. All authors have read and agreed to the published version of the manuscript.

**Funding:** Financial support was provided by the DFG via the GRK (Research Training Group) 2204 "Substitute Materials for Sustainable Energy Technologies."

**Data Availability Statement:** The WANS data from Grenoble presented in this study are openly available at https://doi.ill.fr/10.5291/ILL-DATA.5-26-218, reference number 10.5291/ILL-DATA.5-26-218, accessed on 17 February 2023. Restrictions apply to the availability of the WAXS data. The WANS data from Berlin was obtained from Torben Pfaff and Felix Badaczewski and are available from the authors.

**Acknowledgments:** The authors thank the ILL for the allocation of neutron radiation beamtime and thankfully acknowledge the financial support from the ILL. The authors also appreciate the support of Dr. Dominique Schüpfer with respect to the Raman spectroscopy data.

**Conflicts of Interest:** The authors declare no conflict of interest.

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
