# Peer review of "On the Highly Ordered Graphene Structure of Non-Graphitic Carbons (NGCs)—A Wide-Angle Neutron Scattering (WANS) Study"

_carbon, 2022_

Round 1

Reviewer 1 Report

Osswald and co-workers have applied wide-angle neutron scattering to probe the structure of highly ordered graphene of non-graphitic carbons. The applied approach (WANS) is very effective to probe the microstructural features of carbonaceous materials, the applied methodology and data analysis are very well done, and the paper is quite informative. This work is a great addition to the field, and I think it should be published as is.  

Author Response

Dear Reviewer,

thanks a lot for your kindly reply!

Best Regards,
Oliver OSswald

Reviewer 2 Report

This article discusses non-graphitisable carbons and compares the own results of the authors with the litterature.
This is particularly relevant and interesting for readers interested by the carbon topic.
Of course, there are some questions with this long article:
1/ p5: there is a discussion about the background but no value is given. We expect for example at the 002 position, the height of the carbon peak, the height of the incoherent background, and so one.
having values is really important to follow and be able to understand the weight of the contributions.
2/ p8 : raman : what is the laser power? The abreviate of minute is min and not mm
3/ For the Scherrer formula (eq.2), the reference 43 is given so we expect to find in this publication why the value 0.93 is used.
This is not the case, so it seems the reference is just self-citation. If Scherrer is not cited, we expect a paper where the choice of this value is explained.
4/ In figure 7 : Lc (500°C) is greater than Lc (800°C), a comment is required to explain is there is a physical/chemical reason to that
5/ For La in figure 10, the differences are really huge, the authors propose some convolution effets, which is credible but in fact, we expect to see a zoom in the 10 range to see if the resolution is sufficiant and how are the data.
6/ For the Raman part, p23, the linewidth of the D band must be given and possibly used to obtain La (see Puech et al, Carbon, 105, 275-281, 2016).
A comment on the symmetry of the 2D band can be also given because it corresponds to a turbostratic stacking (see Cancado et al, Carbon, 46(2), 272-275, 2008)

Author Response

Dear reviewer,

thanks a lot for your comments! In short, we added all your suggestions in the manuscript revision:

1 / p5: In Figure 3, additional absolute values for the intensities are given. Hence, the contributions of the incoherent background and the coherent scattering can be better identified.

2 / p8: The laser power was added.

3 / p10: An explanation for choosing a value of 0.93 was added. Generally, this factor lies in the range of 0.9 for fundamental reasons. Since the determination of crystallite dimensions using the Scherrer equation is afflicted with substantial uncertainties anyway, it does not really matter in our context which of the different values is used.

4 / p14: The decrease in Lc at 500 °C might be related to the pyrolysis of gases during the final phase of the coking process, which coincides with previous studies, see ref. 37. We added corresponding text in the caption of figure 7 as well as in the body text (section 3.2.1).

5 / p18: For MP 2100, a zoomed picture was added as Figure S15 in the SI file. It is seen that the 10 interference is reasonably fitted in all cases. However, the huge differences in the La values (Fig. 10 D) do not result from insufficient fitting of data at such low s values, but stem from the range of large s values, because at large s the different effects of finite size and strain/disorder become strongly apparent. 

6 / p23: We agree that the Raman spectroscopy part might be extended in order to provide a comprehensive comparison of the results from Raman spectroscopy with WANS. Indeed, the linewidth of the D band might be used to estimate La. However, we prefer keeping this part short, for several reasons:
1.    As already mentioned in the manuscript, a detailed validation of the D- and G-band analysis against wide-angle x-ray scattering on the same (=identical) carbon materials has already been performed in our previous work (Ref. 25, Schüpfer et al., Carbon 2020, 161, 359–372, doi:10.1016/j.carbon.2019.12.094). Hence, there is no need to show all of the data/analysis again in the present study. We tried to minimize this part. 
2.    In the work of Schüpfer et al. (Ref. 25) it is demonstrated and explained, based on interpreting Raman spectroscopy in terms of the band structure and phonon dispersion, that the usage of the linewidth of the D band is delicate and may be unsuitable, in particular for small La values < 6nm. For instance, the linewidth of the D (defect) band can have strong contributions not only from La, but also sig1. We do not like to enter such detailed discussion in our manuscript and refer to our previous publications.
Having thoroughly considered the reviewer’s suggestion, we now decided to remove Fig. 13E (FWHM of G-band): one should either show both, D- and G-band linewidths, or leave them out both of them. The present manuscript should be regarded as starting point for re-analysing these Raman spectroscopy data, thus a profound further analysis of the linewidths should be part of a separate study.

A comment on the symmetry of the 2D band can be also given because it corresponds to a turbostratic stacking (see Cancado et al, Carbon, 46(2), 272-275, 2008)
See our comment of the previous comment: the 2D bands were discussed in the previous publication (ref. 25). Hence, we added at statement in the text that the reader is referred to ref. 25 regarding the detailed analysis of Raman spectroscopy of these materials.

Best regards,

Oliver Osswald

Round 2

Reviewer 2 Report

The article has been improved and the authors have taken into account reasonably my comments. So, the manuscript is now publishable without any other corrections.